# Questionnaires of interoception do not assess the same construct

Luca Vig [1,2]*, Ferenc Köteles [2], Eszter Ferentzi [2]

1 Doctoral School of Psychology, ELTE Eötvös Loránd University, Budapest, Hungary, 2 Institute of Health Promotion and Sport Sciences, ELTE Eötvös Loránd University, Budapest, Hungary

* vig.luca@ppk.elte.hu

**Data Availability Statement:** All relevant data are within the paper and its Supporting Information files. (The dataset can be found in an online repository under https://osf.io/35ng2/.)

**Funding:** This research was supported by the Hungarian National Scientific Research Fund [K

## Abstract

There are a number of questionnaires assessing the self-reported trait-like aspect of interoception, also called interoceptive sensibility (ISb). Based on the varying purposes of their development and characteristics, however, it is not likely that they assess exactly the same construct. In a community sample of 265 adults, we examined this assumption for three commonly used questionnaires of ISb, namely the Body Awareness subscale of the Body Perception Questionnaire (BPQ-BA), the Body Awareness Questionnaire (BAQ), and the eight subscales of Multidimensional Assessment of Interoceptive Awareness (MAIA). We investigated their associations, and their relation to positive and negative affect and somatosensory amplification. According to the results of correlation analysis, BPQ-BA, BAQ and MAIA were partly unrelated to each other, partly showed weak to moderate positive associations. Also, differences with respect to their association with positive and negative affect were found. These findings suggest that the investigated questionnaires cannot be used interchangeably to assess the subjective aspect of interoception, and the term ISb is not appropriately defined.

## Introduction

Interoception is defined as the sense of the physiological condition of the body [1] current approaches also include the interpretation and integration of bodily signals on both conscious and unconscious levels of processing [2]. Empirical findings indicate the importance of interoception concerning affect, cognitive processes, and various mental health conditions [2, 3]. To develop a unified terminology for various aspects of interoception, Garfinkel and colleagues [4] proposed a model that has become widely used. In this framework three dimensions are described: 1) interoceptive sensibility (ISb), the self-reported or "subjective" aspect, 2) interoceptive accuracy, the "objective" aspect, as assessed by sensory-behavioral measurements, and 3) interoceptive awareness, i.e., the correspondence between the actual and perceived performance in an interoceptive task. Interoceptive accuracy and ISb are usually found to be independent of each other [4–7].

ISb was originally defined by Garfinkel and colleagues as "the self-perceived dispositional tendency to be internally self-focused and interoceptively cognisant" (4, p. 67). Since then,

124132]. Website: https://nkfih.gov.hu/for-the-applicants. The funders had no role in study design, data collection and analysis, decision to publish, or preparation of the manuscript.

**Competing interests:** The authors have declared that no competing interests exist.

further efforts were made to refine the taxonomy of interoception-related constructs. For example, Khalsa and colleagues [2] introduced a "self-report scales" category to the nomenclature of interoception which includes all psychometric state or trait type assessments; also, they pointed out the necessity of further investigation of this category. More recently, Suksasilp and Garfinkel [8] suggested a novel 8-dimension model, in which "self-report and interoceptive beliefs" form one dimension that explicitly covers ISb. This dimension comprises both questionnaires and prior beliefs understood in the Bayesian predictive coding framework; and there are additional dimensions assessed with questionnaires of interoception.

In this paper, the term ISb refers to the trait-like constructs measured by interoception questionnaires. We hope to contribute to the ongoing work of clarification by taking a closer look at the concept of ISb as well as its assessment.

Questionnaires assessing ISb aim to measure beliefs about one's perception of the internal states and changes of their own body. As mentioned, ISb is considered a trait-like characteristic [2, 4, 9]; the completion of questionnaires assessing the construct requires generalization over a broad time span, for which the activation and evaluation of the autobiographical memory are needed [2]. Also, ISb can be regarded as a generalization over a number of various interoceptive modalities, such as heartbeat, gastric sensations, and respiration [2].

In the last decades, several questionnaires have been developed to measure constructs that appear closely related to ISb as defined by Garfinkel and colleagues [4]. Although the implicit assumption is that these questionnaires assess basically the same (or at least highly similar aspects of the same) phenomenon, after inspecting the items of various questionnaires, one has the impression that the underlying concepts may show considerable differences in some important aspects.

First, the adaptivity/maladaptivity of body focused attention is a frequently discussed question in the literature of interoception [10, 11]. One dominant approach highlights the therapeutic, potentially beneficial effects of body focus, e.g., in relation to mind-body practices and various types of psychotherapy [12–17]. For example, the Body Responsiveness Questionnaire [13] and the majority of the subscales of the Multidimensional Assessment of Interoceptive Awareness (MAIA) [18] are designed from this perspective. These constructs usually show a positive association with positive affect (see below). In contrast, the "dark side" of body focus, associated with negative affect and pathological perception of the body, as in health anxiety/hypochondriasis and somatization tendency, is also known [19–22]. For example, the Somatosensory Amplification Scale [23] measures the tendency to experience a bodily sensation as intense, noxious, and disturbing. It is important to note that, although body focus/vigilance is an important aspect of somatosensory amplification, the construct is usually not put under the umbrella of ISb. Another example of a clinically relevant measure of ISb is the Interoceptive Sensitivity and Attention Questionnaire (ISAQ) [24], that differentiates healthy controls and different patient groups by three factors (sensitivity to neutral bodily sensations, attention to unpleasant bodily sensations, and difficulty disengaging from unpleasant bodily sensations).

Second, nearly all the questionnaires assessing interoception work with a uni-dimensional approach [10]. Mehling and colleagues [18] argued, however, that a multidimensional construct is more suitable to grasp all the therapeutically relevant features of ISb. The development of the MAIA was based on this view [18, 25]. In line with the assumed multidimensionality, the use of the total score is not recommended for the MAIA [26].

Third, there are substantial differences among various ISb-related constructs with respect to the extent of cognitive processing of bodily sensations. On the one hand, some measures primarily or exclusively assess the tendency to notice bodily changes and to be aware of bodily processes (e.g. the Body Awareness Subscale of the Body Perception Questionnaire, BPQ-BA [27]). This belongs to the experiential aspect of conscious awareness, also called primary

consciousness [28–30]. On the other hand, various constructs also include higher levels of cognitive processing (aka secondary consciousness). For instance, attention intentionally focused on bodily sensations in order to regulate internal states or to make decisions are often part of ISb-related concepts [10]. Also, cognitive evaluation of possible health-related consequences of body sensations and the behavioral steps necessary to avoid a negative outcome or reach a positive outcome might be included in the construct as well [12, 31, 32].

Keeping in mind the aforementioned aspects, now we will examine the three most frequently used questionnaires of ISb [33]: the BPQ-BA [27], the Body Awareness Questionnaire (BAQ) [34], and the MAIA [18]. After, we will outline the results of existing research regarding their relationship with each other, and also the data that reflects on their affective-evaluative background.

The BPQ-BA by Porges [27] is a very common reference as a questionnaire of interoception in the recent literature (e.g. [4, 35–41]) although clearly exteroceptive signals are also included (e.g. "During most situations I am aware of noises associated with my digestion"). Furthermore, there is an item that seems to assess non-interoceptive cognitive functioning ("During most situations, I am aware of difficulty in focusing"). Unfortunately, there is no publication available regarding the development and/or validation of the original 45-item version of the questionnaire [10] which makes its evaluation from the viewpoint of ISb difficult, even questionable according to some authors [29, 42]. Cabrera and colleagues [43] developed and validated the 26-item and 12-item versions of the BPQ-BA (BPQ-BA-26 and BPQ-BA-12, respectively). Here, the construct assessed by the questionnaire is defined as „the subjective experience of information arising from within the body" ([43], p. 1). The majority of the related items refer to sensations that are primarily linked to stress (e.g. "muscle tension in my face" and "palms sweating"). Thus, negative evaluation appears to be necessarily included in the construct [29]. It was also argued [44, 45] that the BPQ-BA is linked to a maladaptive interoceptive attention style that is associated with somatization, hypochondriasis, and anxiety. In line with these theoretical considerations, the BPQ-BA-26 score was found to be positively associated with stress-reactivity and somatosensory amplification, which was regarded as evidence of convergent validity [43]. Also, in the Chinese validation paper of the BPQ-BA-26 [46], its positive relationship with somatization and somatic symptoms of depression was taken as an indicator of convergent validity. Further research showed that higher scores on the BPQ-BA were associated with worse sleep quality in individuals with clinical depression and anxiety [35], Tourette syndrome [39], risk-aversion [47], comorbid depression [48] alexithymia [49] and symptom reporting [50, 51]. These theoretical considerations and empirical findings indicate that this construct has a considerable overlap with negative affect, which questions the applicability of the BPQ-BA for the assessment of ISb defined in a way that does not include affective evaluation [29]. The questionnaire assesses a uni-dimensional construct. Respondents are asked to rate their awareness of each sensation as they perceive them during everyday situations, hence the construct includes the mere tendency to perceive interoceptive signals, and not the higher-level cognitions about them.

The authors of BAQ aimed to develop a measure that assesses dispositional awareness of normal, non-emotive bodily processes [34]. Typical items of the BAQ are "I notice distinct body reactions when I am fatigued" and "I am aware of a cycle in my activity level throughout the day". The discriminant validity of the BAQ was demonstrated by the overall pattern of independence from symptom reporting tendency and its psychological correlates, specifically anxiety, low self-esteem, and neuroticism [34]. The questionnaire is widely used; it was found to be associated with well-being and positive affect in several studies [13, 16, 52–55]. BAQ, as used nowadays, is uni-dimensional, and assesses the tendency to recognize spontaneous bodily sensations; intentionally directed attention is not involved in any of the items. However, the

construct includes higher-level cognition, for example, the prediction of future bodily states too (e.g. "I can accurately predict what time of day lack of sleep will catch up with me").

Finally, the MAIA (18, revised version: 25) was developed to provide clarity on the construct of "interoceptive (body) awareness" and differentiate between adaptive and maladaptive aspects of self-reported bodily focus. The convergent and divergent validity were tested in several studies and negative associations between trait anxiety and the MAIA subscales were found. This suggests that the particular phenomena that are measured by the MAIA are „clearly not positively related to anxiety or anxiety-associated hypervigilance" ([44], p.3). The MAIA is multidimensional; however, in a recent study that evaluated the psychometric properties of the questionnaire on a large sample [9], six of the eight subscales (Noticing, Attention Regulation, Emotional Awareness, Self-Regulation, Body Listening, and Trusting) were strongly related to a single underlying factor (dubbed MAIA-g factor), which was closely associated (r = .58, p < .001) with the BAQ; the remaining two subscales (Not-Distracting, Not-Worrying; both strongly related to negative affect), were largely independent of the MAIA-g factor. This questions the claimed multidimensionality of the assessed construct; in fact, it supports the classic adaptive *vs* maladaptive distinction (see above) rather than the existence of multiple independent aspects [9]. Concerning the level of cognitive processing, only the Noticing subscale refers to the direct body experience; according to Mehling [44], it is the most similar to the earlier questionnaires (and concepts) of bodily awareness. All the other dimensions are related to higher-level cognitive functions, such as intentional and purposive attentional processes (e.g. "I can refocus my attention from thinking to sensing my body", subscales: Attention Regulation, Not-Distracting, Body Listening, Self-Regulation), and/or the appraisal of the somatic experience (e.g. "I feel my body is a safe place", subscales: Not-Worrying, Emotional Awareness, Trusting) [18].

In a recently published study conducted on a large community sample [33], correlational analysis showed positive associations between the total score of the most cited questionnaires, including MAIA, BAQ and BPQ-BA. The BAQ and MAIA were strongly related, whereas the BPQ-BA was moderately associated with the BAQ and the MAIA. Based on these findings and the overall factorial structure and the network structure of these questionnaires, it was concluded that they measure different constructs. In another recent paper [50] the total scores of the three questionnaires were used, and similar positive associations were found. Again, as the meaning of the MAIA total score is unclear (see above), some MAIA-related findings in these two studies are difficult to interpret. In the latter paper, six of the eight MAIA subscales showed a significant positive association with both the BAQ and the BPQ-BA-12 (12-item version of the BPQ-BA) score, whereas Not-Distracting and Not-Worrying did not. The Polish validation of the MAIA reported a similar relationship between the MAIA subscales and BAQ in a sample of women practicing fitness [56]. In another current study that worked with a large sample [9], the so-called MAIA-g factor (consisting of six subscales out of eight, see above) was strongly related to the BAQ scores, while the Not-Distracting and Not-Worrying subscales showed weak to medium association with it. Pearson and Pfeifer [57] found that the BPQ-BA-45 (45-item version of BPQ-BA) score did not have a significant relationship with five of the MAIA subscales, whilst it showed a moderate positive association with Noticing, a weak positive association with Emotional Awareness and a weak negative association with the Trusting subscale. Furthermore, whereas the MAIA Noticing scale showed a weak positive relationship and the BPQ-BA score showed a medium positive association with neuroticism, the Attention regulation (weak association), the Self-regulation (medium association) and the Trusting (strong association) subscales of the MAIA were negatively related to it. Similar associations were found in another study: the Noticing, Not-Worrying, Attention Regulation, Self-regulation, Body Listening and Trusting MAIA subscales were negatively related to

emotionality, a trait that strongly overlaps with neuroticism [9]. In a meta-analysis on the association of interoception and alexithymia [49], it was reported that 10 out of the reviewed 16 studies used either BPQ-BA or MAIA, and interestingly while the score of BPQ-BA had a positive association with alexithymia, the Noticing and the Emotional Awareness subscales of MAIA were reversely related to it. Concerning positive affect, a positive association with the BAQ was found (see above). Also, the majority of the MAIA subscales showed weak to moderate positive associations with extraversion, which is closely related to positive affect [9].

In summary, theoretical considerations and empirical evidence do not convincingly support the notion that the three questionnaires (BPQ-BA, BAQ, MAIA) assess the same construct, i.e., ISb, and the assumption that they are interchangeable. In the cross-sectional questionnaire study presented in this paper, we intended to shed more light on the associations between these questionnaires, as well as on their evaluative background. More specifically, we tested the following hypotheses:

1. Relationship of the different ISb-related measures: We expected a moderate association between the BAQ, the BPQ-BA, and the eight MAIA subscales.

2. Relationship between certain ISb measures and positive affect: We expected a positive association between positive affect and the BAQ, and six aforementioned MAIA subscales, i.e., Noticing, Attention Regulation, Emotional Awareness, Self-Regulation, Body Listening, and Trusting.

3. Relationship between certain ISb measures and negative affect: We expected that the MAIA Not-Worrying, MAIA Attention Regulation, MAIA Self-regulation, and the MAIA Trusting subscales have a negative, furthermore the BPQ-BA have a positive association with negative affect. Also, a positive association between BPQ-BA and somatosensory amplification was hypothesized.

## Materials and methods

### Participants

The data collection was conducted between July and September 2020 in Hungarian language. This study is part of a larger longitudinal study, data from the first measurement were used. The online survey was advertised on the web page of a Hungarian Psychology themed magazine. Participants did not receive any payment; but they could receive feedback on their scores of the eight MAIA subscales and their explanation based on the original publication of the questionnaire [18] following the data collection period if they requested. Originally, 392 individuals started to fill out the questionnaire; 127 of them were excluded because they quit before completing the entire test battery. Finally, answers of 265 participants were used (age = 38.2 ±11.45 yrs, 84% female) in this study. In terms of educational qualification, 78,5% reported higher education (university degree), 20,8% secondary level education (high school), and less than 1% primary level education. The study was approved by the Ethical Board of the University (Approval Nr. 2020/289); all participants signed an online informed consent form.

### Questionnaires

The Body Awareness Questionnaire (BAQ) [34] measures beliefs about the perceived sensibility to normal (i.e. non-pathological), non-emotive bodily processes. The assessment has an emphasis on the sensitivity to bodily cycles and rhythms, the ability to detect small, normal bodily changes caused by fatigue, hunger, lack of sleep, etc., and the ability to anticipate bodily

reactions. The validated Hungarian version of the BAQ consists of 17 statements, rated on a 7-point Likert scale (1 = not at all true about me, 7 = very true about me) [53]. Higher scores represent a higher level of body awareness. The internal consistency of the questionnaire (McDonald's omega) in this study was 0.823.

The 26-item Body Awareness Subscale of the short form of Body Perception Questionnaire (BPQ-BA-26) [43] measures the sensibility to anxiety-related bodily processes. The original 122-item version of the Body Perception Questionnaire was developed by Porges [27] in order to measure body awareness, stress response, autonomic nervous system reactivity, stress style, and health history. The short form of BPQ only includes the Body Awareness (26 items) and Autonomic Reactivity subscales (20 items), The items are rated on a 5-point Likert scale (1 = Never, 5 = Always). The first Hungarian translation of the 26-item short-form of the Body Awareness subscale is presented in this article. The English version was translated by two researchers independently. After they agreed on the final form together with a third expert, a fourth independent person translated it back to English. The original and the back-translated English versions were compared by a native speaker. The McDonald's omega in the present study was 0.970. Data on the factor structure of the Hungarian version of the questionnaire are available in the S1 File.

The Multidimensional Assessment of Interoceptive Awareness (MAIA) measures various self-reported aspects of interoception [9, 18]. It consists of 32 items across 8 subscales: Noticing, Not-Distracting, Not-Worrying, Attention Regulation, Emotional Awareness, Self-Regulation, Body Listening, and Trusting. Items are rated on a 5-point Likert scale (1 = Never, 5 = Always). The internal consistency (McDonald's omega coefficients) of the subscales was between 0.687 and .879 (Noticing: 0.731; Not-Distracting: 0.687; Not-Worrying: 0.767; Attention Regulation: 0.863; Emotional Awareness: 0.825; Self-Regulation: 0.825; Body Listening: 0.800, Trusting: 0.879).

The short version of the Positive and Negative Affect Schedule (I-PANAS-SF) [58, 59] consists of two subscales: trait-like positive affect (PA; containing a variety of pleasant mood states e.g., enthusiasm), and negative affect (NA; containing aversive mood states e.g., nervousness); each measured by 5 items. Participants have to rate on a five-point Likert scale, how much each statement describes how they usually feel (1 = Very slightly or not at all to 5 = Very much). The McDonald's omega coefficient for the positive affect scale was .769 and that for the negative affect scale was 0.761.

The Somatosensory Amplification Scale (SSAS) [23, 60] assesses the individual tendency to experience a somatic sensation as intense, noxious, and disturbing. The questionnaire consists of 10 items covering uncomfortable sensations that are usually not related to serious illness (e.g. "Even something minor, like an insect bite or a splinter, really bothers me"). Participants have to rate on a 5-point Likert scale how much they agree with each statement (1 = not at all; 5 = extremely). The internal consistency (McDonald's omega coefficient) in this study was 0.668.

## Statistical analysis

Statistical analysis was conducted using the JASP v0.14.3 software [61]. Associations between the assessed variables were estimated using Spearman correlation due to the violation of normality for a number of variables. As the overall number of correlation analyses testing our hypotheses was high, the accepted level of significance was set to $p < .001$. Strength of the associations were interpreted following the recommendation of Cohen for the social sciences, i.e., r = 0.1 was regarded as small, r = 0.3 as medium, and r = 0.5 as large effect size [62]. Linear regression analysis was used to examine variables' individual contribution to positive and

**Table 1. Descriptive statistics of the assessed variables.**

|  | N | M | SD | minimum | maximum |
|---|---|---|---|---|---|
| BPQ-BA-26 | 265 | 84.53 | 26.23 | 31 | 130 |
| BAQ | 226 | 85.07 | 12.16 | 48 | 119 |
| MAIA Noticing | 233 | 3.75 | 0.81 | 1 | 5 |
| MAIA Not-Distracting | 233 | 3.04 | 0.85 | 1 | 5 |
| MAIA Not-Worrying | 233 | 2.93 | 0.95 | 1 | 4.67 |
| MAIA Attention Regulation | 233 | 3.4 | 0.79 | 1 | 5 |
| MAIA Emotional Awareness | 233 | 3.88 | 0.86 | 1 | 5 |
| MAIA Self-Regulation | 233 | 3.12 | 0.89 | 1 | 5 |
| MAIA Body Listening | 233 | 3.2 | 0.96 | 1 | 5 |
| MAIA Trusting | 233 | 3.83 | 0.88 | 1 | 5 |
| SSAS | 225 | 31.34 | 5.71 | 17 | 47 |
| Negative affect | 258 | 10.72 | 3.57 | 5 | 21 |
| Positive affect | 258 | 18.52 | 3.23 | 5 | 25 |

Note. BPQ-BA-26 = Body Awareness Subscale of the short form of Body Perception Questionnaire; BAQ = Body Awareness Questionnaire; MAIA = Multidimensional Assessment of Interoceptive Awareness; SSAS = Somatosensory Amplification Scale

negative affect. All variables were entered in the equation in one step, using the ENTER method. The dataset can be found in an online repository under https://osf.io/35ng2/.

## Results

Descriptive statistics are presented in Table 1, the output of the correlation analysis testing the expected associations between BPQ-BA-26, BAQ and the MAIA subscales (Hypothesis 1) is presented in Table 2 and Fig 1. The correlation between the BPQ-BA-26 and BAQ is $r_s$ = .27, p < .001. The BAQ showed weak to medium strength associations with six MAIA subscales but not with MAIA Not-Distracting and MAIA Not-Worrying, whereas the BPQ-BA-26 was weakly associated with MAIA Noticing, MAIA Not-Distracting and MAIA Body Listening.

Concerning Hypothesis 2, positive affect showed a positive association with BAQ ($r_s$ = .34, p < .001), MAIA Noticing ($r_s$ = .38, p < .001), MAIA Attention Regulation ($r_s$ = .34, p < .001), MAIA Emotional Awareness ($r_s$ = .28, p < .001), MAIA Self-Regulation ($r_s$ = .26, p < .001),

**Table 2. Spearman correlations between the assessed measures of ISb (the association between BPQ-BA-26 and BAQ is $r_s$ = .27, p < .001).**

|  | BPQ-BA-26 | BAQ |
|---|---|---|
| MAIA Noticing | .24*** | .42*** |
| MAIA Not-Distracting | .24*** | .13 |
| MAIA Not-Worrying | -.05 | .09 |
| MAIA Attention Regulation | .17 | .27*** |
| MAIA Emotional Awareness | .17 | .41*** |
| MAIA Self-Regulation | .14 | .24*** |
| MAIA Body Listening | .23*** | .37*** |
| MAIA Trusting | .13 | .22*** |

Note. BPQ-BA-26 = Body Awareness Subscale of the short form of Body Perception Questionnaire; BAQ = Body Awareness Questionnaire; MAIA = Multidimensional Assessment of Interoceptive Awareness

***: p < .001, i.e., the accepted level of p

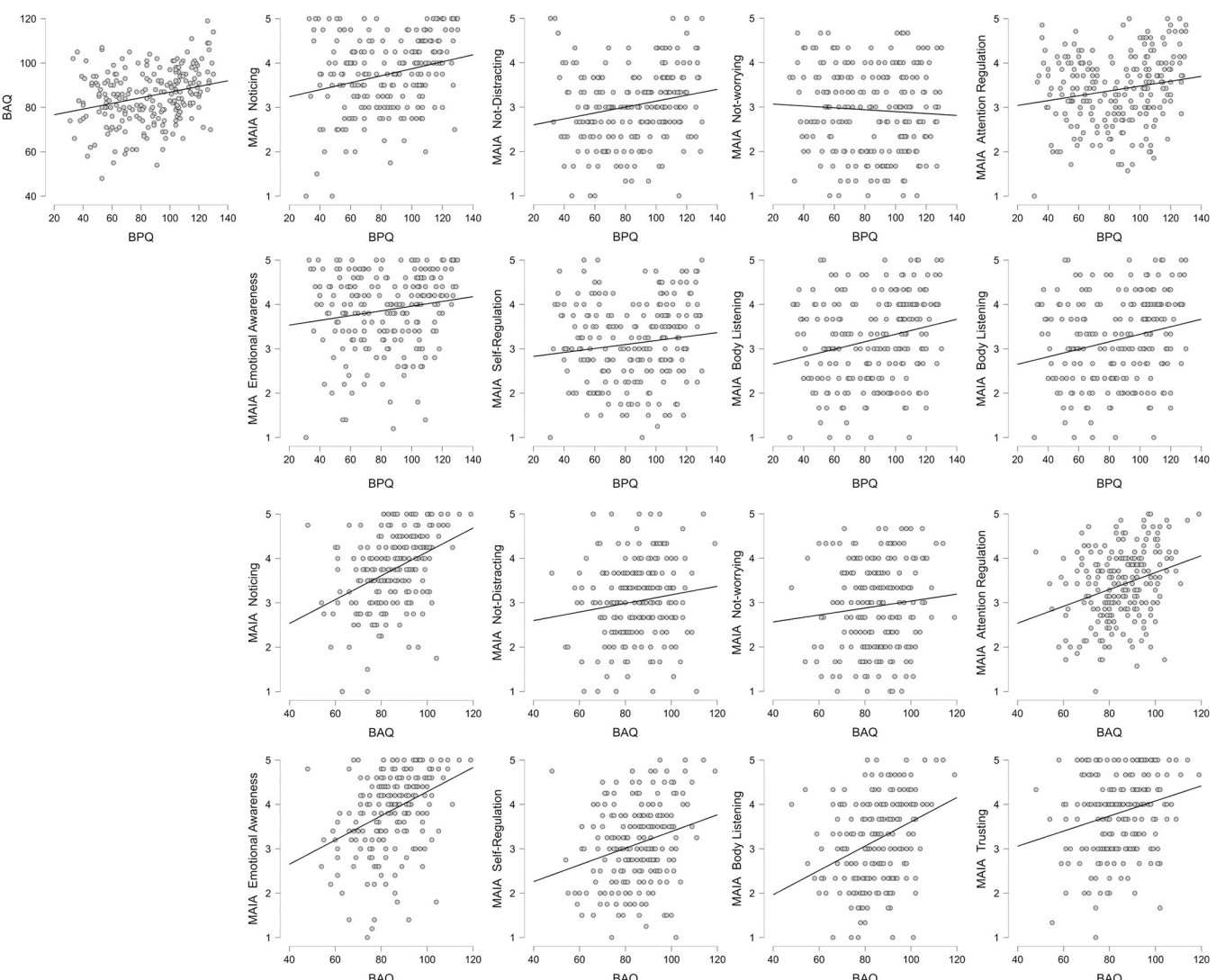

**Fig 1. Scatter plots presenting the pairwise associations between the assessed questionnaires of ISb.** BPQ = Body Awareness Subscale of the short form of Body Perception Questionnaire; BAQ = Body Awareness Questionnaire; MAIA = Multidimensional Assessment of Interoceptive Awareness.

MAIA Body Listening ($r_s$ = .26, p < .001), and MAIA Trusting ($r_s$ = .29, p < .001). In contrast, the associations between positive affect and MAIA Not-Distracting ($r_s$ = .03, p = .690), MAIA Not-worrying ($r_s$ = .14, p = .027), and the BPQ-BA-26 ($r_s$ = .12, p = .048) were non-significant. The linear regression analysis with positive affect as criterion variable and the 7 scales that showed a significant association with positive affect (see above) indicated the significant individual contribution of MAIA_Noticing, MAIA_Trusting and the BAQ ($R^2$ = 0.196, *p* < 0.001; for details, see Table 3).

Finally (Hypothesis 3), negative affect was not associated with BPQ-BA-26 ($r_s$ = .14, p = .028), MAIA Noticing ($r_s$ = -.12, p = .071), MAIA Not-Distracting ($r_s$ = -.04, p = .585), MAIA Emotional Awareness ($r_s$ = -.07, p = .313), MAIA Body Listening ($r_s$ = -.15, p = .020), and the BAQ ($r_s$ = -.18, p = .006). However, it showed a negative association with MAIA Not-worrying ($r_s$ = -.31, p < .001), MAIA Attention Regulation ($r_s$ = -.24, p < .001), MAIA Self-Regulation ($r_s$ = -.28, p < .001), and MAIA Trusting ($r_s$ = -.37, p < .001). The linear regression analysis

**Table 3. Results of multiple linear regression analysis with positive affect as criterion variable.**

| Variable | Unstandardized B | SE of B | Standardized β | p |
|---|---|---|---|---|
| (Intercept) | 8.414 | 1.558 | | < .001 |
| MAIA_Noticing | 0.741 | 0.337 | 0.186 | 0.029 |
| MAIA_Attention Regulation | 0.426 | 0.356 | 0.104 | 0.233 |
| MAIA_Emotional Awareness | 0.028 | 0.335 | 0.007 | 0.934 |
| MAIA_Self-Regulation | 0.008 | 0.296 | 0.002 | 0.978 |
| MAIA_Body Listening | -0.065 | 0.285 | -0.019 | 0.821 |
| MAIA_Trusting | 0.582 | 0.285 | 0.157 | 0.042 |
| BAQ | 0.044 | 0.018 | 0.165 | 0.016 |

Note: MAIA = Multidimensional Assessment of Interoceptive Awareness; BAQ = Body Awareness Questionnaire

with negative affect as criterion variable and the latter 4 MAIA subscales as predictors indicated that MAIA Not-worrying and MAIA Trusting showed a significant negative association ($R^2$ = 0.169, $p$ < 0.001; for details, see Table 4). The BPQ-BA-26 was weakly associated with SSAS ($r_s$ = .24, p < .001).

Additionally, the assessed variables were subjected to exploratory factor analysis (for details, see S2 File). Two weakly connected factors were revealed. Factor 1 was characterized by positive affect, BAQ, MAIA Noticing, MAIA Attention Regulation, MAIA Emotional Awareness, MAIA Self-Regulation, MAIA Body Listening, and MAIA Trusting. Factor 2 was associated with negative affect, SSAS and MAIA Not-worrying (reverse association). The BPQ-BA-26 showed a moderate positive association with both factors.

## Discussion

The main goal of our study was to examine the associations between the most cited questionnaires of ISb and further explore their affective-evaluative background. In a community sample consisting of 265 adults, self-report questionnaires of ISb (i.e. the BPQ-BA-26, the BAQ, and the eight subscales of MAIA) were partly unrelated to each other, partly showed positive associations.

This finding only partly supports our first hypothesis. In more detail, the BAQ was weakly to moderately associated with the BPQ-BA-26 and six MAIA subscales (Noticing, Attention Regulation, Emotional Awareness, Self-Regulation, Body Listening, and Trusting), whereas the BPQ-BA-26 was weakly related to three MAIA subscales only (Noticing, Not-Distracting, Body Listening). The difference between our results and the above-cited findings of Gajdos and colleagues [50] and Pearson and Pfeifer [57] regarding the associations between the subscales of the MAIA and the BPQ could be explained by the characteristics of the samples.

Also, the BAQ and the aforementioned six MAIA subscales showed weak to moderate positive associations with positive affect, supporting our second hypothesis. Two MAIA subscales,

**Table 4. Results of multiple linear regression analysis with negative affect as criterion variable.**

| Variable | Unstandardized B | SE of B | Standardized β | p |
|---|---|---|---|---|
| (Intercept) | 17.423 | 1.150 | | < 0.001 |
| MAIA_Not-worrying | -0.842 | 0.243 | -0.223 | < 0.001 |
| MAIA_Attention Regulation | 0.312 | 0.352 | 0.069 | 0.375 |
| MAIA_Self-Regulation | -0.379 | 0.314 | -0.094 | 0.228 |
| MAIA_Trusting | -1.090 | 0.305 | -0.267 | < 0.001 |

Note: MAIA = Multidimensional Assessment of Interoceptive Awareness

namely Noticing, and Trusting and the BAQ showed independent contribution to positive affect. Noticing refers to the awareness (the frequency of noticing) of comfortable, neutral, and uncomfortable bodily sensations. Its independent contribution can be explained in at least two ways. One is reversed: a better emotional state frees up attentional capacity, allowing the individual to allocate more attentional resources to various stimuli, including physical sensations [52]. Another interpretation—in line with the approach which highlights the adaptivity of mind-body practices—is that having information from the body contributes to well-being, at least when it is functioning properly (e.g. [10]). The contribution of Trusting is understandable as it shows the extent of experiencing one's own body as safe and trustworthy. This result is in line with the findings of Hanley and colleagues [26]; in their study Trusting showed the highest zero-order correlation with well-being among the MAIA subscales. The BAQ, besides measuring the tendency to notice small changes in the body, also assesses the awareness of bodily cycles and the ability to anticipate bodily reactions. These skills can add to a sense of control which can explain the association with positive affect. To sum up, direct experience of and trust in the body and the sense of predictability seem to have importance when it comes to the association between positive affect and interoceptive sensibility, which aspects are measured by two MAIA subscales and the BAQ.

In accordance with our third hypothesis, four MAIA subscales (Not-worrying, Attention Regulation, Self-Regulation, Trusting) were reversely associated with negative affect. Out of these dimensions, Not-worrying and Trusting had an individual contribution to negative affect (in the reverse direction). Consistent with our results, construct validity examinations found that these two subscales (and Attention-Regulation) have the strongest negative association with trait anxiety-related measures [44]. Not-Worrying refers to the tendency of not reacting to pain or physical discomfort with emotional distress and worry, i.e., a general proneness to reduce negative affective states. Trusting can decrease symptoms of anxiety and depression by diminishing the extent of uncertainty [26]. Unexpectedly, the BPQ-BA-26 was independent of both positive and negative affectivity; on the other hand, it had a positive association with the anxiety-related somatosensory amplification, as presumed.

Based on the commonly used three-dimensional model of interoception [4] the aforementioned three constructs are often considered more or less equivalent. In other words, although this is not explicitly stated, ISb is not only conceptualized as a term covering different self-reported measures of interoception but (as it is assessed with questionnaires) also as a unitary construct with various, possibly equivalent ways to operationalize. Although this issue has been highlighted by a number of authors (e.g. [2, 33, 44]), papers citing Garfinkel and colleagues [4] often build on the assumption of a unitary construct [63, 64]. Based on our results, there is no reason to suppose the interchangeability of the investigated questionnaires; thus, the usage of the term ISb for questionnaires of interoception is not well justified. It gives the misleading impression that these measures are interchangeable. Most importantly, we found the association between the BPQ-BA-26 and the other measures of interoception weak or non-significant, while the BAQ showed weak to moderate associations with most of the questionnaires included in our study. There were also important differences among the scales with respect to their association with positive and negative affect. These findings, however, are not at all surprising if we take a closer look at the constructs behind the questionnaires.

## Further comments on adaptivity and emotional evaluation

Adaptivity, i.e., the positive or negative outcome of being sensitive and attentive to bodily signals, is strongly linked to the way individuals evaluate and interpret perceived body sensations However, we do not want to suggest that negative affect associated with some of them is not

useful and necessary for healthy functioning. It is well-known that certain important body percepts, such as pain, hunger for air (breathlessness), and the tension of the stomach, are unpleasant, i.e., they are subject to automatic negative evaluation [29, 65]. As described in the introduction, however, interoception measured with questionnaires is not or only weakly related to the ability to perceive peripheral physiological changes; in fact, affect appears to impact more the individual's belief about their body focus and the perceived ability to notice somatic changes (this is what questionnaires asses) than the majority of bottom-up signals originating in the body. Negative evaluation often leads to the interpretation of perceived body signals as symptoms that indicate actual or possible pathology [22, 66, 67], as in the case of somatosensory amplification [68]. In contrast, positive evaluation more easily leads to an adaptive interpretation, i.e., it makes the individual explain perceived interoceptive information as helpful and/or an indicator of healthy functioning. Of course, causation is not uni-directional in either case. Perceived symptoms can further increase (health) anxiety which in turn leads to introspection, prediction, and active seeking of signs of pathology [19, 20].

Keeping these considerations in mind, the BPQ-BA appears to be quite balanced. In contrast, the BAQ and the MAIA subscales are more related to positive affect. However, the BPQ-BA is more negatively biased concerning the involved bodily percepts, as the vast majority of the items cover bodily signs of sympathetic activity and stress [29]. Hence, its association with somatosensory amplification is not surprising.

## Dimensionality and the level of cognitive processing

Authors of the MAIA [18] work with a multidimensional approach. In accordance with this, the associations between the subscales of the MAIA (r ranging from .09 to .60) are interpreted as indicators of independence [18], and usually no total score is calculated and used [26]. As the subscales of the MAIA do not measure the same construct, they necessarily show different associations with other constructs. The other two measures included in this study, i.e., the BPQ-BA-26 and the BAQ, represent a unidimensional approach in their recently used form and are only weakly associated.

Whereas the BPQ-BA and the MAIA Noticing subscale assess the spontaneous tendency to sense or notice interoceptive sensations, the MAIA and the BAQ largely involve secondary evaluation (i.e. the meaning of the percepts), the purposeful direction of attention, and prediction. In our opinion, it is important that the core of the ISb construct, as defined by Garfinkel and colleagues [4] includes only the pure detection and primary perception of interoceptive signals. From this point of view, the BPQ-BA and MAIA Noticing subscale appear to be more suitable to measure ISb than the other MAIA subscales and BAQ. It is important to note that the issue of ineffability may also impact the self-report of body sensations; as they belong to primary consciousness, language often lacks the terms to describe them [69–71]. Thus, individual differences in the ability to translate such sensations into words may substantially impact the self-reports, regardless of their actual frequency [29].

## Remarks on the phrasing of the questionnaires

Beyond the above-mentioned ineffability issue (or related to them), the phrasing of the questionnaires is another problematic point. For instance, the BPQ-BA asks participants to rate their awareness of each of the described bodily states or processes. One should judge the prevalence of being aware of the listed sensations „during most situations". This brings up the question that if the respondent only rarely experiences certain sensations, how can it be decided whether these bodily processes occur often but mostly remain unnoticed (low score) or they occur rarely and are usually noticed (high score). Some participants of our study gave us

feedback on this controversial phrasing of the BPQ-BA. While the statements of the BPQ-BA refer to very specific sensations, the BAQ focuses more on the context of the bodily sensations, but not the frequency of the sensations themselves (e.g. "I notice distinct body reactions when I am fatigued."). The MAIA works with the most general description of the bodily states (e.g. "I notice where in my body I am comfortable"). However, certain items of MAIA might be more understandable for mind-body practiced individuals than for those without such history, as it was developed with instructors and practitioners of these methods [18]. These language-related characteristics of the questionnaires can also affect the responses.

## On the usability of the questionnaires

Based on the aforementioned considerations, it can be concluded that none of the investigated measures can be regarded as the optimal questionnaire of the self-reported trait-like aspect of interoception. On the one hand, the definition of ISb provided by Garfinkel and colleagues [4] is very broad (see e.g. "self-evaluated assessment of subjective interoception, gauged using interviews/questionnaires", p.65), necessarily including a comparatively wide variety of constructs. On the other hand, another important characteristic (and at the same time limitation) of the entire framework of Garfinkel et al. [4] is that it focuses on the sensory aspect of interoception and largely ignores the evaluative aspect [29, 32, 72].

As the questionnaires included in this study do not measure the same construct, authors of future studies should carefully consider the distinctive features of the individual measures in order to choose the scale that best suits their needs. To assess the perceived ability to sense signals originating from within the body (i.e. primary percepts), the MAIA Noticing appears to be the best option. The BPQ-BA also measures direct experience, but its emphasis on sympathetic activation-related sensations makes it more appropriate for the investigation of the subjective aspects of stress. The BAQ might be the primary choice to explore the background of the association between body awareness and positive affect, including the direction of causality. Finally, as the majority of the MAIA subscales include affective and/or cognitive evaluation (e.g. meaning) of the perceived sensations, they appear usable in studies that focus on the evaluative aspect of interoception.

## Limitations

Firstly, our study investigated only three out of the available questionnaires of interoception. Although these are the most popular ones and theoretically represent a wide range of the possible approaches, there are other options, such as the Autonomic Perception Questionnaire [73], the Private Body Consciousness Scale [74] and the Scale of Body Connection [75], just to mention a few. Secondly, to understand better the constructs assessed by the investigated questionnaires, the involvement of a wider range of variables is desirable, in accordance with the multitrait, multimethod framework [76]. According to this, besides convergent validation (the involvement of variables that possibly assess something similar), discrimination validity is also a significant aspect of the salivation process. Thus, during the critical examination of the existing questionnaires, it could be also informative, whether they really differ from the questionnaires they ought to differ from (e.g. measures of mindfulness); this was only partially fulfilled in our recent study. Third, a non-representative community sample was used which limits the overall generalizability of the findings. Fourth, the internal consistency of the SSAS was quite low. Cronbach's alpha values in this domain are commonly reported in the literature and might reflect the conceptual heterogeneity of the construct (for a detailed discussion, see [68]). Fifth, cultural and linguistic factors might also limit the generalizability of the findings (for discussion see [77]). Finally, the sample size is relatively small for exploratory factors analysis.

## Conclusion

Questionnaires of interoception cannot be used interchangeably. Available empirical evidence on the associations between ISb and other, partly pathological constructs should be reconsidered depending on the questionnaire(s) used in different studies. Thus, one has to keep in mind that results obtained with a certain questionnaire of interoception are specific and cannot be generalized to other questionnaires. This is an important aspect to consider both during the planning of the study (in order to choose the right measure) and when the results are interpreted (in order to avoid over-generalization).

## Supporting information

**S1 File. Factor structure of the BPQ.**
(PDF)

**S2 File. Exploratory factor analysis of the assessed variables.**
(PDF)

## Author Contributions

**Conceptualization:** Luca Vig, Ferenc Köteles, Eszter Ferentzi.

**Data curation:** Luca Vig.

**Formal analysis:** Ferenc Köteles.

**Methodology:** Eszter Ferentzi.

**Supervision:** Ferenc Köteles, Eszter Ferentzi.

**Visualization:** Ferenc Köteles.

**Writing – original draft:** Luca Vig.

**Writing – review & editing:** Luca Vig, Ferenc Köteles, Eszter Ferentzi.

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
