## [Decision Letter · Decision Letter 0]

16 Jul 2021

PONE-D-21-13357

Questionnaires of interoception do not assess the same construct

PLOS ONE

Dear Dr. Vig,

Thank you for submitting your manuscript to PLOS ONE. After careful consideration, we feel that it has merit but does not fully meet PLOS ONE’s publication criteria as it currently stands. Therefore, we invite you to submit a revised version of the manuscript that addresses the points raised during the review process.

We look forward to receiving your revised manuscript.

Kind regards,

Delphine Grynberg, PhD

Academic Editor

PLOS ONE

Journal Requirements:

Reviewers' comments:

Reviewer's Responses to Questions

**Comments to the Author**

1. Is the manuscript technically sound, and do the data support the conclusions?

Reviewer #1: Partly

Reviewer #2: Yes

Reviewer #3: Partly

2. Has the statistical analysis been performed appropriately and rigorously? 

Reviewer #1: No

Reviewer #2: Yes

Reviewer #3: No

3. Have the authors made all data underlying the findings in their manuscript fully available?

Reviewer #1: No

Reviewer #2: Yes

Reviewer #3: Yes

4. Is the manuscript presented in an intelligible fashion and written in standard English?

Reviewer #1: Yes

Reviewer #2: Yes

Reviewer #3: Yes

5. Review Comments to the Author

Reviewer #1: I would like to commend the authors on a well-written manuscript. The introduction sets up contrasts between these questionnaires well. The widespread use of interoceptive sensibility as a research construct makes the understanding of convergence between measures important.

However, I have some concerns about the analysis that was conducted in this study.

1. Information on distributions of the subscale scores are not reported. From the scatterplots it appears that there may be some skew in the measures, which is particularly apparent in the MAIA Noticing subscale. Non-normality can create artifacts in the parametric statistics that are used in this analysis. I would recommend assessing normality and using linear transformation if needed to ensure alignment between distributions and model assumptions

2. The sample here is relatively small for exploratory factor analysis. I would recommend that the authors consider the role of sample size and its implications for the success of uncovering the underlying factor structure. These articles may provide useful background in this area:

Floyd, F. J. & Widaman, K. F. (1995). Factor analysis in the development and refinement of clinical assessment instruments. Psychological Assessment, 7(3), 286-299.

MacCallum, R. C., Widaman, K. F., Zhang, S., & Hong, S. (1999). Sample size in factor analysis. Psychological Methods, 4(1), 84-99.

3. The decision of how many factors to retain is crucial for this study. In the manuscript, factor retention was solely guided by the eigenvalue > 1 rule, which is known to have limitations. Typical factor analysis recommendations now include comparing fit indices of each solution, the scree test, and/or parallel analysis. Since the key outcome of this study rests on retaining the correct amount of factors, it is important that multiple metrics are used to guide factor retention. The following article provides an overview on this topic:

Fabrigar, L. R., Wegener, D. T., MacCallum, R. C., & Strahan, E. J. (1999). Evaluating the use of exploratory factor analysis in psychological research. Psychological Methods, 4, 272-299.

4. There is no mention of whether oblique or orthogonal rotation was used (based on results, I assume oblique)

5. I could not find supporting data in the supplemental file

6. Data collection and participant features are only briefly described. Several clarifications would be needed: Is country of residence data on participants available? Were there any cleaning procedures to screen out potentially poor quality or incomplete responses? Also, based on the questionnaire descriptions it is assumed that the survey was in Hungarian but this could also be useful in the participant description.

Reviewer #2: The current study examines associations between three interoceptive sensibility measures, and positive and negative affect by means of correlations and exploratory factor analysis in a cross-sectional community sample of 265 adults. I would like to present the following questions and suggestions to the authors.

Abstract

• In the abstract, the SSAS abbreviation is used without introduction.

Introduction

• L.44: To my opinion, the statement that interoceptive accuracy and sensibility are usually independent needs to be formulated a bit more cautiously. As it is formulated now, the statement would imply that interoceptive awareness is usually nonexistent, which is not the case. It is not because generally low correlations are found between interoceptive accuracy and sensibility (for existing measures across participants), that both can be considered independent.

• L.58: I am a bit confused by the use of the term “pure perception”. Perception, and interoception specifically, involves the sensing, identification, differentiation and interpretation of internal sensations, and implies by definition potential bias by associated emotions, cognitions and beliefs, for example deriving from personality factors including self-esteem and affect, as suggested by the authors.

• L.63: It is not completely clear what the “total score” of questionnaires refers to precisely. I assume that the authors refer to the overall underlying construct assessed by questionnaires?

• L.136 and later (continuing in the discussion): If I understand correctly, the authors argue that moderate to high correlations (e.g. between specific interoception questionnaires and positive and negative affect) imply a conceptual overlap between both. Although a correlation represents statistical overlap in terms of variance explained, it does not necessarily imply a conceptual overlap, however. At most, it may suggest a conceptual overlap.

• L.146-151: To me, it is not clear what the precise goal is of this paragraph. Two domains are mentioned in which the BAQ is commonly used. However, the BAQ is used in much more research domains. Therefore it is not clear what the precise message here is. If the two research domains mentioned are examples, this could be made more explicit.

• Throughout the introduction, the authors repeatedly use the term “susceptibility”. This is not a term commonly used in the interoception literature, and to me it is not clear how it is defined by the authors. Based on what the authors write, but I may misunderstand, it seems they mean mostly “sensitivity” in this regard? Given that “susceptibility” means often more than “sensitivity” and may also imply vulnerability, the term may not be the best choice. In any case, a clear definition is required, especially in the interoception literature in which constructs are generally poorly delineated, or at least have a history of that.

• I would slightly disagree with the authors that the BPQ, BAQ and MAIA are currently used interchangeably by researchers. All three questionnaires are developed with their own purpose and outline the concept they are measuring in different ways. Often this concept is specified by the researchers choosing a specific questionnaire, and the choice is accounted for. However, I do agree with the statement that often all these different questionnaires are considered measuring interoception, which is fundamentally different than them being considered and used interchangeable. Researchers who aim to measure interoceptive sensibility often choose a questionnaire of the three questionnaires used here, but that does not mean they consider them being exchangeable. I do want to add that this does not take away that the message that they should not be used interchangeable (as suggested in the conclusion), is an important message.

Methods

• L.225: It is not clear to me what is meant with “feedback upon request”.

• L.2226: Maybe the editor could address this, but I am not sure whether the information provided on ethics approval is sufficient for PLOS ONE.

• L.232: Since the Hungarian translation of the BAQ seems to differ from the original BAQ in number of items, it may be good to add a reference (I assume Köteles, 2014) directly to this sentence to show a validated version was used.

• L. 254: I wonder what the rationale is to average scores of the MAIA g-factor scales? The average (vs. the sum) limits the range a lot.

• Table 1: The minimum score for the SSAS is probably a type error?

• Is any more information available to better describe the community sample which was investigated? Also, more information on the recruitment process (e.g. how/for which purpose was the study advertised) may be helpful to better understand the studied sample.

Results:

• It would be helpful to add the correlations in the text as well, so it becomes more clear how the strength of the correlations is interpreted. In this regard, it would also be helpful if the authors could indicate which norms or boundaries they used to assess the strength of correlations.

o For example, a correlation of 0.251 (BPQ-BA-26 and MAIA noticing) is considered weak, whereas a correlation of 0.269 (BPQ-BA-26 and SSAS) is considered weak to moderate. Or a correlation of -0.168 (BAQ – NA) is considered weak whereas a correlation of 0.154 (BPQ-BA-26 – NA) is interpreted as very weak.

o In addition, the SSAS is mentioned to only correlate with NA and BPQ-BA-26 in the results, however also shows a correlation of 0.143 with the BAQ (which is higher than the correlation between PA and BPQ-BA-26, which is interpreted as very weak); this correlation is not discussed in the results, yet elaborated upon in the discussion.

o Also, it seems that the associations between BAQ and MAIA noticing/-g are not discussed in the results, as is the association between MAIA noticing and PA, yet both relationships are elaborated upon in the discussion.

o In the results, the associations between BPQ-BA-26 and BAQ are described as weak to moderate, while in the discussion the relationship is described as particularly weak.

• Same goes for the interpretations of the factor loadings. Which cutoff scores were used here? For example a factor loading of 0.3 was interpreted as the BPQ-BA-26 not being part of factor 1, whereas a factor loading of 0.337 was considered indicative of the SSAS belonging to factor 2.

Discussion:

• It is not clear to me, based on the predictions made in the introduction, why the authors describe the moderate to strong relationship between MAIA-g and BAQ (in the results described as moderate) to be remarkable.

• The discussion of NA and its role in interoception is particularly long (l.376-412), and the link with the current findings only comes after. As a reader I felt a bit lost in this part as I did not know where the story was going.

• In this discussion, also the formulation of a questionnaire being “positively or negatively biased”, is unclear to me. To my opinion, based on the current research design and findings, the most accurate conclusion would be in terms of associations.

Minor suggestions:

l.223, l. 272: data were

l.229: The Body Awareness Questionnaire

l.249: The MAIA does not consist of 32 items of 8 scales. It consists of 32 items across 8 scales.

Reviewer #3: This manuscript addresses an important question. However, I have a few concerns that need to be addressed.

ABSTRACT

Typo? On should in In

Final sentence is not clear – is the conclusion that we need a new questionnaire?

INTRODUCTION

Both the question and the concepts are clearly and accurately defined in this section. My main concern is the length – it is over seven pages long. I would recommend shortening this section.

Rational – some decisions seem arbitrary

(1) the choice of questionnaires needs justification. Why were these three questionnaires chosen over others? E.g., the IAS Murphy, J., Brewer, R., Plans, D., Khalsa, S. S., Catmur, C., & Bird, G. (2020). Testing the independence of self-reported interoceptive accuracy and attention. Quarterly Journal of Experimental Psychology, 73(1), 115-133, or the SCS Fenigstein A, Scheier MF, Buss AH (1975) Public and private self-consciousness: Assessment and theory. J Consult Clin Psychol 43 (4) 522–527 , or the BMQ Burg, J. M., Probst, T., Heidenreich, T., & Michalak, J. (2017). Development and psychometric evaluation of the body mindfulness questionnaire. Mindfulness, 8(3), 807-818. I’m sure there are others……………..

(2) Additionally, what was the reason for focusing on the noticing and ‘g’ scales of the MAIA? You mention in the introduction that the emotional awareness scale had previously been associated with the other questionnaire ……… yet you chose not to focus on the individual scales? This needs more justification. The ‘g’ scale seems to confound a range of different interoceptive constructs. I would recommend considering the scale separately – afterall, the aim of the present paper is to highlight to researchers that such differences need to be considered?

(3) My understanding of The Somatosensory Amplification Scale is that previous research has found it lacks internal reliability. One potential reason is that it may measure more than one construct. As teasing apart body related constructs was the aim of the present paper why use this scale to compare the others against?

ANALYSIS

It might be problematic to add the noticing scale of the MAIA and ‘g’ together into the factor analysis when ‘g’ already contains the noticing scale………again this might be a reason to include each of the MAIA scales separately. Additionally, should the individual items, rather than the scales, go into the factor analysis?

DISCUSSION

The discussion is again very long and reads like a review of questionnaire measures rather than a discussion of the findings. While it raises a number of interesting points the focus should be on the present results.

I am not sure I agree with the statements about an ideal interoception questionnaire L 459-464. Surely there is no such thing as an ‘ideal’ questionnaire, rather researchers should carefully select their questionnaire based on their research question and the underlying interoceptive construct they want to measure. I am not sure it makes conceptual sense to entirely separate interoception and affect as suggested in point 2 in this section. I am also not sure that an ‘ideal’ questionnaire would exclude the evaluative component – surely this is an important component, and again, whether it is assessed will depend on the research question.

6. PLOS authors have the option to publish the peer review history of their article (what does this mean?). If published, this will include your full peer review and any attached files.

Reviewer #1: No

Reviewer #2: No

Reviewer #3: No

---

## [Author Response · Author response to Decision Letter 0]

14 Sep 2021

Dear Reviewers,

Many thanks for your suggestions and comments on our manuscript. All issues you raised were carefully addressed in the revised version (see below). In our opinion, these changes considerably improved the overall quality of the manuscript. Hopefully, you will find the revised version appropriate for publication.

Best wishes,

Luca Vig,

corresponding author

Reviewer #1: I would like to commend the authors on a well-written manuscript. The introduction sets up contrasts between these questionnaires well. The widespread use of interoceptive sensibility as a research construct makes the understanding of convergence between measures important.

Reply 1: Many thanks for the positive evaluation. We also think that this is an important area that has received relatively little attention in the past.

However, I have some concerns about the analysis that was conducted in this study.

1. Information on distributions of the subscale scores are not reported. From the scatterplots it appears that there may be some skew in the measures, which is particularly apparent in the MAIA Noticing subscale. Non-normality can create artifacts in the parametric statistics that are used in this analysis. I would recommend assessing normality and using linear transformation if needed to ensure alignment between distributions and model assumptions

Reply 2: We decided to use Pearson correlation because of its robustness (e.g. Havlicek & Peterson, 1976; Knief & Forstmeier, 2021). In the revised version, this was changed to Spearman correlation (this change did not substantially impact the outcome of the analysis).

References:

Havlicek, L. L., & Peterson, N. L. (1976). Robustness of the Pearson Correlation against Violations of Assumptions. Perceptual and Motor Skills, 43(3_suppl), 1319–1334. https://doi.org/10.2466/pms.1976.43.3f.1319

Knief, U., & Forstmeier, W. (2021). Violating the normality assumption may be the lesser of two evils. Behavior Research Methods. https://doi.org/10.3758/s13428-021-01587-5

____

2. The sample here is relatively small for exploratory factor analysis. I would recommend that the authors consider the role of sample size and its implications for the success of uncovering the underlying factor structure. These articles may provide useful background in this area:

Floyd, F. J. & Widaman, K. F. (1995). Factor analysis in the development and refinement of clinical assessment instruments. Psychological Assessment, 7(3), 286-299.

MacCallum, R. C., Widaman, K. F., Zhang, S., & Hong, S. (1999). Sample size in factor analysis. Psychological Methods, 4(1), 84-99.

Reply 3: Determination of minimum required sample size is a highly debated issue in the context of exploratory factor analysis. Certain authors recommend a minimum sample size of n = 100; also, the minimum acceptable ratio of number of participants to number of variables varies from 3 to 20 (for a review, see Mundfrom et al., 2005). As the sample size is above n = 200 and the participants to variables ratio is above 25 in our study, the data met these criteria. Still, as the bigger the better with respect to desirable sample size for EFA, this is a limitation of the study, which is mentioned in the revised version.

Reference:

Mundfrom, D. J., Shaw, D. G., & Ke, T. L. (2005). Minimum Sample Size Recommendations for Conducting Factor Analyses. International Journal of Testing, 5(2), 159–168. https://doi.org/10.1207/s15327574ijt0502_4

____

3. The decision of how many factors to retain is crucial for this study. In the manuscript, factor retention was solely guided by the eigenvalue > 1 rule, which is known to have limitations. Typical factor analysis recommendations now include comparing fit indices of each solution, the scree test, and/or parallel analysis. Since the key outcome of this study rests on retaining the correct amount of factors, it is important that multiple metrics are used to guide factor retention. The following article provides an overview on this topic:

Fabrigar, L. R., Wegener, D. T., MacCallum, R. C., & Strahan, E. J. (1999). Evaluating the use of exploratory factor analysis in psychological research. Psychological Methods, 4, 272-299.

Reply 4: Many thanks for this remark. In the revised version, parallel analysis was used to determine the number of factors to be extracted (it also indicated 2 factors thus does not change the analysis).

____

4. There is no mention of whether oblique or orthogonal rotation was used (based on results, I assume oblique)

Reply 5: In the original version, the promax rotation with Kaiser normalization and κ = 4 in SPSS was used which is an oblique rotation. In the revised version, factor analysis was conducted with JASP, also with oblique rotation (these details are explicitly described in the revised version; see also Reply 36).

5. I could not find supporting data in the supplemental file

Reply 6: A supplemental file will be uploaded with the revised version.

____

6. Data collection and participant features are only briefly described. Several clarifications would be needed: Is country of residence data on participants available? Were there any cleaning procedures to screen out potentially poor quality or incomplete responses? Also, based on the questionnaire descriptions it is assumed that the survey was in Hungarian but this could also be useful in the participant description.

Reply 7: The Participants section of the revised version was completed with these pieces of information

____

Reviewer #2: The current study examines associations between three interoceptive sensibility measures, and positive and negative affect by means of correlations and exploratory factor analysis in a cross-sectional community sample of 265 adults. I would like to present the following questions and suggestions to the authors.

Abstract

• In the abstract, the SSAS abbreviation is used without introduction.

Reply 8: This is corrected in the revised version

____

Introduction

• L.44: To my opinion, the statement that interoceptive accuracy and sensibility are usually independent needs to be formulated a bit more cautiously. As it is formulated now, the statement would imply that interoceptive awareness is usually nonexistent, which is not the case. It is not because generally low correlations are found between interoceptive accuracy and sensibility (for existing measures across participants), that both can be considered independent.

Reply 9: Interoceptive awareness refers to the association between actual and perceived (typically measured with VAS) performance in a modality-specific sensory-behavioral task (e.g. a heartbeat perception/detection task) and indeed shows individual differences with respect to the strength of the association. However, ISb, i.e., a self-reported trait-like characteristic assessed with a questionnaire, usually shows no association with interoceptive accuracy.

____

• L.58: I am a bit confused by the use of the term “pure perception”. Perception, and interoception specifically, involves the sensing, identification, differentiation and interpretation of internal sensations, and implies by definition potential bias by associated emotions, cognitions and beliefs, for example deriving from personality factors including self-esteem and affect, as suggested by the authors.

Reply 10: The distinction between primary (experiential) and secondary (language-based) consciousness and processes is widely used (although admittedly not perfect). Concerning interoception, this distinction is used to differentiate between automatic (associative) and reasoning-based evaluative processes. For example, Farb and Logie (2018) make this distinction with respect to interoceptive appraisal, whereas Herbert and Pollatos state (2018) that interoceptive affective evaluation refers to the former category only. Nevertheless, the term “pure perception” was removed from the revised version and a concise description of primary and secondary consciousness is provided (“This belongs to the experiential aspect of conscious awareness, also called primary consciousness (Farthing, 1991; Köteles, 2021; Lloyd, 1989). On the other hand, various constructs also include higher levels of cognitive processing (aka secondary consciousness). For instance, attention intentionally focused on bodily sensations in order to regulate internal states or to make decisions are often part of the various ISb-related concepts (Mehling et al., 2009).”).

References:

Herbert, B. M., & Pollatos, O. (2018). The relevance of interoception for eating behavior and eating disorders. In M. Tsakiris & H. De Preester (Eds.), The interoceptive mind. From homeostasis to awareness (pp. 165–186). Oxford University Press.

Farb, N. A., & Logie, K. (2018). Interoceptive appraisal and mental health. In M. Tsakiris & H. De Preester (Eds.), The interoceptive mind. From homeostasis to awareness (pp. 227–241). Oxford University Press.

____

• L.63: It is not completely clear what the “total score” of questionnaires refers to precisely. I assume that the authors refer to the overall underlying construct assessed by questionnaires?

Reply 11: Thank you for this remark. The use of the term in that context was inappropriate, we removed it (“Also, ISb can be regarded as a generalization over a number of various interoceptive modalities, such as heartbeat, gastric sensations, and respiration (Khalsa et al., 2018)”).

____

• L.136 and later (continuing in the discussion): If I understand correctly, the authors argue that moderate to high correlations (e.g. between specific interoception questionnaires and positive and negative affect) imply a conceptual overlap between both. Although a correlation represents statistical overlap in terms of variance explained, it does not necessarily imply a conceptual overlap, however. At most, it may suggest a conceptual overlap.

Reply 12: This statement was reconsidered and reworded as follows: “These results suggest that the investigated measures of ISb might assess different (although to some extent overlapping) constructs.”

____

• L.146-151: To me, it is not clear what the precise goal is of this paragraph. Two domains are mentioned in which the BAQ is commonly used. However, the BAQ is used in much more research domains. Therefore it is not clear what the precise message here is. If the two research domains mentioned are examples, this could be made more explicit.

Reply 13: The paragraph is reworded in the revised version; it describes the BAQ from a more relevant point of view (“The questionnaire is widely used; it was found to be associated with well-being and positive affect in a number of studies (Daubenmier, 2005; Impett et al., 2006; Moradi & Huang, 2008; Tihanyi, Böőr, et al., 2016)”).

____

• Throughout the introduction, the authors repeatedly use the term “susceptibility”. This is not a term commonly used in the interoception literature, and to me it is not clear how it is defined by the authors. Based on what the authors write, but I may misunderstand, it seems they mean mostly “sensitivity” in this regard? Given that “susceptibility” means often more than “sensitivity” and may also imply vulnerability, the term may not be the best choice. In any case, a clear definition is required, especially in the interoception literature in which constructs are generally poorly delineated, or at least have a history of that.

Reply 14: The term was removed from the revised version.

____

• I would slightly disagree with the authors that the BPQ, BAQ and MAIA are currently used interchangeably by researchers. All three questionnaires are developed with their own purpose and outline the concept they are measuring in different ways. Often this concept is specified by the researchers choosing a specific questionnaire, and the choice is accounted for. However, I do agree with the statement that often all these different questionnaires are considered measuring interoception, which is fundamentally different than them being considered and used interchangeable. Researchers who aim to measure interoceptive sensibility often choose a questionnaire of the three questionnaires used here, but that does not mean they consider them being exchangeable. I do want to add that this does not take away that the message that they should not be used interchangeable (as suggested in the conclusion), is an important message.

Reply 15: This part was reworded as follows: “The recently used categorization of the dimensions of interoception suggests that the described dimensions are relatively homogeneous (Garfinkel, Seth, et al., 2015), thus questionnaire measures of ISb are interchangeable, i.e., they measure the same construct. In other words, ISb is not only conceptualized as a term covering different self-reported measures of interoception but (as it is assessed with questionnaires) also as a unitary construct (i.e. dimension) with various, possibly equivalent ways to operationalize. Although this is not explicitly stated and rarely investigated, papers citing Garfinkel and colleagues (2015) often build on this assumption (Eng et al., 2020; Mul et al., 2018). Based on our results, there is no reason to suppose the interchangeability of the three investigated questionnaires for the measurement of ISb.”

____

Methods

• L.225: It is not clear to me what is meant with “feedback upon request”.

Reply 16: The sentence was modified: “Participants could receive feedback on their scores of the eight MAIA scales and it’s explanation based on the original publication of the questionnaire (Mehling et al., 2012) following the data collection period if they requested.”

____

• L.2226: Maybe the editor could address this, but I am not sure whether the information provided on ethics approval is sufficient for PLOS ONE.

Reply 17: Approval number is provided in the revised version. Also, it is stated that all participants signed an informed consent form.

____

• L.232: Since the Hungarian translation of the BAQ seems to differ from the original BAQ in number of items, it may be good to add a reference (I assume Köteles, 2014) directly to this sentence to show a validated version was used.

Reply 18: Reference added.

____

• L. 254: I wonder what the rationale is to average scores of the MAIA g-factor scales? The average (vs. the sum) limits the range a lot.

Reply 19: We followed the original scoring instructions (averaging might have been used to make the subscale scores directly comparable). However, averaging (i.e. a linear transformation of the scores) is not problematic from a statistical point of view in the present correlational study.

____

• Table 1: The minimum score for the SSAS is probably a type error?

Reply 20: Corrected, thank you for spotting it.

____

• Is any more information available to better describe the community sample which was investigated? Also, more information on the recruitment process (e.g. how/for which purpose was the study advertised) may be helpful to better understand the studied sample.

Reply 21: Information about the recruitment process is provided in the revised version.

____

Results:

• It would be helpful to add the correlations in the text as well, so it becomes more clear how the strength of the correlations is interpreted. In this regard, it would also be helpful if the authors could indicate which norms or boundaries they used to assess the strength of correlations.

o For example, a correlation of 0.251 (BPQ-BA-26 and MAIA noticing) is considered weak, whereas a correlation of 0.269 (BPQ-BA-26 and SSAS) is considered weak to moderate. Or a correlation of -0.168 (BAQ – NA) is considered weak whereas a correlation of 0.154 (BPQ-BA-26 – NA) is interpreted as very weak.

Reply 22: In the revised version, the “classic” recommendations of Cohen (0.1: weak, 0.3: medium, 0.5: strong) were consistently used and this is stated in the statistical analysis section. Correlations were added to the text.

____

o In addition, the SSAS is mentioned to only correlate with NA and BPQ-BA-26 in the results, however also shows a correlation of 0.143 with the BAQ (which is higher than the correlation between PA and BPQ-BA-26, which is interpreted as very weak); this correlation is not discussed in the results, yet elaborated upon in the discussion.

Reply 23: In contrast to Pearson correlation, Spearman correlation coefficient was non-significant. In consequence, this topic was removed from the discussion.

____

o Also, it seems that the associations between BAQ and MAIA noticing/-g are not discussed in the results, as is the association between MAIA noticing and PA, yet both relationships are elaborated upon in the discussion.

Reply 24: These associations are mentioned in the results section of the revised version.

____

o In the results, the associations between BPQ-BA-26 and BAQ are described as weak to moderate, while in the discussion the relationship is described as particularly weak.

Reply 25: Particularly was omitted in the revised version (“The association between the BPQ-BA-26 and the other measures of ISb is weak”)

____

• Same goes for the interpretations of the factor loadings. Which cutoff scores were used here? For example a factor loading of 0.3 was interpreted as the BPQ-BA-26 not being part of factor 1, whereas a factor loading of 0.337 was considered indicative of the SSAS belonging to factor 2.

Reply 26: In the revised version, loadings above .3 were considered only.

____

Discussion:

• It is not clear to me, based on the predictions made in the introduction, why the authors describe the moderate to strong relationship between MAIA-g and BAQ (in the results described as moderate) to be remarkable.

Reply 27: This sentence was reworded in the revised version (“Findings indicate a moderate association between the MAIA-g and the BAQ, which suggest an overlap between the two underlying constructs (Ferentzi et al., 2020).”)

____

• The discussion of NA and its role in interoception is particularly long (l.376-412), and the link with the current findings only comes after. As a reader I felt a bit lost in this part as I did not know where the story was going.

Reply 28: This part of the discussion was substantially shortened.

____

• In this discussion, also the formulation of a questionnaire being “positively or negatively biased”, is unclear to me. To my opinion, based on the current research design and findings, the most accurate conclusion would be in terms of associations.

Reply 29: The term bias was removed from these parts of the discussion.

____

Minor suggestions:

l.223, l. 272: data were

l.229: The Body Awareness Questionnaire

l.249: The MAIA does not consist of 32 items of 8 scales. It consists of 32 items across 8 scales.

Reply 30: Corrected.

____

Reviewer #3: This manuscript addresses an important question. However, I have a few concerns that need to be addressed.

ABSTRACT

Typo? On should in In

Reply 31: Corrected.

____

Final sentence is not clear – is the conclusion that we need a new questionnaire?

Reply 32: This is the second (more distal) part of the conclusion, which is preceded by a more direct conclusion (“These findings suggest that the investigated questionnaires cannot be used interchangeably to assess interoceptive sensibility.”). At the end of the discussion of the manuscript, we conclude that the existing questionnaires do have their weaknesses, thus there is room for another, theoretically well-founded questionnaire.

____

INTRODUCTION

Both the question and the concepts are clearly and accurately defined in this section. My main concern is the length – it is over seven pages long. I would recommend shortening this section.

Reply 33: The introduction was substantially shortened.

____

Rational – some decisions seem arbitrary

(1) the choice of questionnaires needs justification. Why were these three questionnaires chosen over others? E.g., the IAS Murphy, J., Brewer, R., Plans, D., Khalsa, S. S., Catmur, C., & Bird, G. (2020). Testing the independence of self-reported interoceptive accuracy and attention. Quarterly Journal of Experimental Psychology, 73(1), 115-133, or the SCS Fenigstein A, Scheier MF, Buss AH (1975) Public and private self-consciousness: Assessment and theory. J Consult Clin Psychol 43 (4) 522–527 , or the BMQ Burg, J. M., Probst, T., Heidenreich, T., & Michalak, J. (2017). Development and psychometric evaluation of the body mindfulness questionnaire. Mindfulness, 8(3), 807-818. I’m sure there are others……………..

Reply 34: The choice was justified in the revised version (and the limited number of included questionnaires is mentioned among the limitations of the study): “Keeping in mind the aforementioned aspects, now we will take a closer look at three widely used constructs that fit the definition of ISb and the questionnaires designed to assess them: the BPQ-BA, the Body Awareness Questionnaire (BAQ) (Shields et al., 1989), and the MAIA. Besides their popularity, we chose these questionnaires because each of them represents a distinct approach to grasp ISb. The BPQ-BA has a unique approach as long as it focuses on relatively simple body sensations, without involving interpretation. The BAQ is an example of questionnaires that aim to assess normal and non-emotive bodily processes. Last but not least, none of the other measures aims to grasp as many aspects of interoception, as the MAIA.”

____

(2) Additionally, what was the reason for focusing on the noticing and ‘g’ scales of the MAIA? You mention in the introduction that the emotional awareness scale had previously been associated with the other questionnaire ……… yet you chose not to focus on the individual scales? This needs more justification. The ‘g’ scale seems to confound a range of different interoceptive constructs. I would recommend considering the scale separately – afterall, the aim of the present paper is to highlight to researchers that such differences need to be considered?

Reply 34: We think that the g-scale does not confound a range of different constructs but represents an overarching factor behind those constructs. In other words, the differentiation of the six subscales is not really justified, as shown by Ferentzi et al., 2020 (“However, a recent paper showed that the most prominent component of the construct can be well grasped with an overreaching factor (called MAIA-g in this paper) that includes six out of eight factors, while the remaining two (Not-Worrying and Not-Distracting) are independent (Ferentzi et al., 2020).”). Thus, the use of the g-scale appears more appropriate than the inclusion of the separate scales. The only exception is the Noticing scale, because it measures primary experience (“Concerning the level of cognitive processing, only the Noticing subscale refers to the direct body experience, and, according to Mehling (2016), it is the most similar to the earlier questionnaires (and concepts) of bodily awareness.”).

____

(3) My understanding of The Somatosensory Amplification Scale is that previous research has found it lacks internal reliability. One potential reason is that it may measure more than one construct. As teasing apart body related constructs was the aim of the present paper why use this scale to compare the others against?

Reply 35: We had two reasons. First, the SSAS was used in the validation process of the BPQ-BA and their positive association was interpreted as indicator of convergent validity. Second, somatosensory amplification is a construct that is very close to ISb (actually, it fits the definition of Garfinkel et al (2015, p.67): “the self-perceived dispositional tendency to be internally self-focused and interoceptively cognisant”; this is explicitly stated in the revised version), but usually does not considered an aspect of it, mainly because of its association with NA. This makes it an excellent anchor point for our study, which attempts to explore the associations between indicators of ISb and PA/NA.

____

ANALYSIS

It might be problematic to add the noticing scale of the MAIA and ‘g’ together into the factor analysis when ‘g’ already contains the noticing scale………again this might be a reason to include each of the MAIA scales separately. Additionally, should the individual items, rather than the scales, go into the factor analysis?

Reply 36: The size of the sample does not allow the inclusion of individual items into the FA. To address the issue with the noticing subscale and the g-scale, we rerun the factor-analysis with a g-scale that does not contain the four items of the Noticing subscale (the correlation between the full and this shortened g-scale is .99).

____

DISCUSSION

The discussion is again very long and reads like a review of questionnaire measures rather than a discussion of the findings. While it raises a number of interesting points the focus should be on the present results.

Reply 37: Similarly to the introduction, the discussion was substantially shortened in the revised version (see also Reply 28).

____

I am not sure I agree with the statements about an ideal interoception questionnaire L 459-464. Surely there is no such thing as an ‘ideal’ questionnaire, rather researchers should carefully select their questionnaire based on their research question and the underlying interoceptive construct they want to measure. I am not sure it makes conceptual sense to entirely separate interoception and affect as suggested in point 2 in this section. I am also not sure that an ‘ideal’ questionnaire would exclude the evaluative component – surely this is an important component, and again, whether it is assessed will depend on the research question.

Reply 38: We think that the entire framework of Garfinkel and colleagues (2015) relies on the sensory aspect of interoception. We formulated our recommendations with keeping this approach in mind. This does not mean that the evaluative aspect is not important or can be completely separated from the sensory. This is elaborated in the discussion of the revised version.

---

## [Decision Letter · Decision Letter 1]

1 Nov 2021

PONE-D-21-13357R1Questionnaires of interoception do not assess the same constructPLOS ONE

Dear Dr. Vig,

Thank you for submitting your manuscript to PLOS ONE. After careful consideration, we feel that it has merit but does not fully meet PLOS ONE’s publication criteria as it currently stands. Therefore, we invite you to submit a revised version of the manuscript that addresses the points raised during the review process.

 I apologize for the delay. You will find below the comments of R3. I globally agree with these and thus kindly invite you to address them in your revised version. In addition to these comments, here are my comments that I also kindly invite you to address. 1.
In the introduction and the method sections, could you provide more information/illustrations about the BAQ (i.e., items) 2.
Please discuss the low internal reliability of the SSAS3.
I agree with R3 about making your main objective clearer: evaluating the associations between the 3 questionnaires to examine their particularities (rather than proposing guidelines to develop a new questionnaire). 4.
I agree with R3 about the different factors the MAIA because you clearly mentioned in the introduction its multi-dimensionality. Removing the two more “maladaptive scales” is a bit surprising and considering each subscale may provide relevant information. I would conduct correlations with all MAIA subscales5.
I would suggest to remove the exploratory analysis if the sample size is too small to make appropriate analyses or to increase the sample size. 6.
The discussion could indeed more specifically focus on the latent constructs of ISb questionnaires/dimensions based on the correlations (the phrasing issues could be addressed in the limit section). You could also for instance add recommendations about which questionnaires/subscales should be considered in future studies depending on the research objective7.
I understand the relevance to include negative affect but please better justify/discuss the implications of the associations between the ISb questionnaires and PA. 8.
Conclusion: a word is missing ‘cannot be..to ISb’

We look forward to receiving your revised manuscript.

Kind regards,

Delphine Grynberg, PhD

Academic Editor

PLOS ONE

Reviewers' comments:

Reviewer's Responses to Questions

**Comments to the Author**

1. If the authors have adequately addressed your comments raised in a previous round of review and you feel that this manuscript is now acceptable for publication, you may indicate that here to bypass the “Comments to the Author” section, enter your conflict of interest statement in the “Confidential to Editor” section, and submit your "Accept" recommendation.

Reviewer #1: All comments have been addressed

Reviewer #3: (No Response)

2. Is the manuscript technically sound, and do the data support the conclusions?

Reviewer #1: Yes

Reviewer #3: Yes

3. Has the statistical analysis been performed appropriately and rigorously? 

Reviewer #1: Yes

Reviewer #3: No

4. Have the authors made all data underlying the findings in their manuscript fully available?

Reviewer #1: Yes

Reviewer #3: Yes

5. Is the manuscript presented in an intelligible fashion and written in standard English?

Reviewer #1: Yes

Reviewer #3: Yes

6. Review Comments to the Author

Reviewer #1: The authors have addressed all of my comments regarding the manuscript. I recommend the paper for publication; however, I strongly suggest that the authors closely check the writing for grammar before going to press. There is also formatting inconsistency in use of decimal points or commas to separate the integer and fractional part of numbers.

Reviewer #3: Thank you for considering my comments. I still have a few remaining concerns

Reviewer #3:

Final sentence is not clear – is the conclusion that we need a new questionnaire?

Reply 32: This is the second (more distal) part of the conclusion, which is preceded by a more direct conclusion (“These findings suggest that the investigated questionnaires cannot be used interchangeably to assess interoceptive sensibility.”). At the end of the discussion of the manuscript, we conclude that the existing questionnaires do have their weaknesses, thus there is room for another, theoretically well-founded questionnaire. ____

I didn’t have any concerns about the conclusion that the questionnaires should not be used interchangeably. You have provided data that support that conclusion. What is not clear is why you review the limitations of each questionnaire and conclude that a new questionnaire is needed. This is going beyond the aim of your study which was to see if the questionnaire measures the same underlying latent construct. If you want to critically review the interoception questionnaires, then this is a different aim and should be a different paper. I maintain that the final sentence of the abstract does not tally up to the aim of the study.

Additionally, what was the reason for focusing on the noticing and ‘g’ scales of the MAIA? You mention in the introduction that the emotional awareness scale had previously been associated with the other questionnaire ……… yet you chose not to focus on the individual scales? This needs more justification. The ‘g’ scale seems to confound a range of different interoceptive constructs. I would recommend considering the scale separately – afterall, the aim of the present paper is to highlight to researchers that such differences need to be considered?

Reply 34: We think that the g-scale does not confound a range of different constructs but represents an overarching factor behind those constructs. In other words, the differentiation of the six subscales is not really justified, as shown by Ferentzi et al., 2020 (“However, a recent paper showed that the most prominent component of the construct can be well grasped with an overreaching factor (called MAIA-g in this paper) that includes six out of eight factors, while the remaining two (Not-Worrying and NotDistracting) are independent (Ferentzi et al., 2020).”). Thus, the use of the g-scale appears more appropriate than the inclusion of the separate scales. The only exception is the Noticing scale, because it measures primary experience (“Concerning the level of cognitive processing, only the Noticing subscale refers to the direct body experience, and, according to Mehling (2016), it is the most similar to the earlier questionnaires (and concepts) of bodily awareness.”). ____

I still disagree with this. There are now plenty of studies that find some sub-scales of the MAIA relate to particular disorders, while others don’t. This implies they are not measuring the same thing. I would like to see a factor analysis in supplementary information using the individual scales.

(3) My understanding of The Somatosensory Amplification Scale is that previous research has found it lacks internal reliability. One potential reason is that it may measure more than one construct. As teasing apart body related constructs was the aim of the present paper why use this scale to compare the others against?

Reply 35: We had two reasons. First, the SSAS was used in the validation process of the BPQ-BA and their positive association was interpreted as indicator of convergent validity. Second, somatosensory amplification is a construct that is very close to ISb (actually, it fits the definition of Garfinkel et al (2015, p.67): “the self-perceived dispositional tendency to be internally self-focused and interoceptively cognisant”; this is explicitly stated in the revised version), but usually does not considered an aspect of it, mainly because of its association with NA. This makes it an excellent anchor point for our study, which attempts to explore the associations between indicators of ISb and PA/NA. ____

This response doesn’t really address my question about reliability?

ANALYSIS It might be problematic to add the noticing scale of the MAIA and ‘g’ together into the factor analysis when ‘g’ already contains the noticing scale………again this might be a reason to include each of the MAIA scales separately. Additionally, should the individual items, rather than the scales, go into the factor analysis?

Reply 36: The size of the sample does not allow the inclusion of individual items into the FA. To address the issue with the noticing subscale and the g-scale, we rerun the factor-analysis with a g-scale that does not contain the four items of the Noticing subscale (the correlation between the full and this shortened g-scale is .99). ____

Irrespective of the correlation it is still inappropriate to the noticing scale and the g scale (including noticing) into the same factor analysis as this would inflate the chances of the two loading onto the same factor. If removing noticing from the g scale does not influence the results then why not report that analysis? It would be more appropriate?

DISCUSSION The discussion is again very long and reads like a review of questionnaire measures rather than a discussion of the findings. While it raises a number of interesting points the focus should be on the present results. Reply 37: Similarly to the introduction, the discussion was substantially shortened in the revised version (see also Reply 28). ____ I am not sure I agree with the statements about an ideal interoception questionnaire L 459-464. Surely there is no such thing as an ‘ideal’ questionnaire, rather researchers should carefully select their questionnaire based on their research question and the underlying interoceptive construct they want to measure. I am not sure it makes conceptual sense to entirely separate interoception and affect as suggested in point 2 in this section. I am also not sure that an ‘ideal’ questionnaire would exclude the evaluative component – surely this is an important component, and again, whether it is assessed will depend on the research question. Reply 38: We think that the entire framework of Garfinkel and colleagues (2015) relies on the sensory aspect of interoception. We formulated our recommendations with keeping this approach in mind. This does not mean that the evaluative aspect is not important or can be completely separated from the sensory. This is elaborated in the discussion of the revised version.

The discussion still seems to go well beyond a discussion of the present results and the aim. Indeed, there are sections where your results are not referred to. In terms of Garfinkels definition, two scales recently developed by Murphy et al. seem to have taken care of that question. The Interoceptive Accuracy Scale and the Interoceptive Attention Scale. Please comment on what you think is needed beyond these two well validated and conceptualised scales.

7. PLOS authors have the option to publish the peer review history of their article (what does this mean?). If published, this will include your full peer review and any attached files.

Reviewer #1: No

Reviewer #3: No

---

## [Author Response · Author response to Decision Letter 1]

16 Dec 2021

Editor’s comments

1. In the introduction and the method sections, could you provide more information/illustrations about the BAQ (i.e., items)

Reply 1: Additional items were included in the introduction; also, description of the BAQ in the methods section was somewhat extended.

2. Please discuss the low internal reliability of the SSAS:

Reply 2: This issue is discussed among the limitations of the revised version (“Fourth, the internal consistency of the SSAS was quite low. Cronbach’s alpha values in this domain are commonly reported in the literature and might reflect the conceptual heterogeneity of the construct (for a detailed discussion, see Köteles & Witthöft, 2017).”).

3. I agree with R3 about making your main objective clearer: evaluating the associations between the 3 questionnaires to examine their particularities (rather than proposing guidelines to develop a new questionnaire).

Reply 3: Guidelines for a novel questionnaire were removed from the revised version.

4. I agree with R3 about the different factors the MAIA because you clearly mentioned in the introduction its multi-dimensionality. Removing the two more “maladaptive scales” is a bit surprising and considering each subscale may provide relevant information. I would conduct correlations with all MAIA subscales

Reply 4: In the revised version, all MAIA scales are included.

5. I would suggest to remove the exploratory analysis if the sample size is too small to make appropriate analyses or to increase the sample size.

Reply 5: As reviewer 3 explicitly asked us to include a modified version of the factor analysis (see Reply 11), we present it as supplementary material. The analysis includes all MAIA subscales but does not include the MAIA_g. As the outcome of this analysis does not differ from that of our original analysis (using the MAIA Noticing and MAIA_g only), we completely removed the original analysis.

6. The discussion could indeed more specifically focus on the latent constructs of ISb questionnaires/dimensions based on the correlations (the phrasing issues could be addressed in the limit section). You could also for instance add recommendations about which questionnaires/subscales should be considered in future studies depending on the research objective

Reply 6: Recommendations for future studies are included (“As the questionnaires included in this study do not measure the same construct, authors of future studies should carefully consider the distinctive features of the individual measures in order to choose the scale that best suits their needs. To assess the perceived ability to sense signals originating from within the body (i.e. primary percepts), the MAIA Noticing appears to be the best option. The BPQ-BA also measures direct experience, but its emphasis on sympathetic activation-related sensations makes it more appropriate for the investigation of the subjective aspects of stress. The BAQ might be the primary choice to explore the background of the association between body awareness and positive affect, including the direction of causality. Finally, as the majority of the MAIA subscales includes affective and/or cognitive evaluation (e.g. meaning) of the perceived sensations, they appear usable in studies that focus on the evaluative aspect of interoception.”).

Concerning the paragraph on the phrasing issues, we think that this is not a limitation of the current study, but a more general issue impacting all kinds of primary percepts. As it is rarely discussed or even mentioned in the recent literature, we feel that is worthy of mentioning.

7. I understand the relevance to include negative affect but please better justify/discuss the implications of the associations between the ISb questionnaires and PA.

Reply 7: The impact and importance of PA is discussed in more detail in the revised version (“With respect to an adaptive interpretation, noticing and appreciating the messages of the body can improve positive affect and subjective well-being. On the other hand, better psychological functioning enables people to allocate more attentional resources to various stimuli, including information originating in the body (Ferentzi et al., 2019).”; “The BAQ might be the primary choice to explore the background of the association between body awareness and positive affect, including the direction of causality.”).

8. Conclusion: a word is missing ‘cannot be..to Isb’

Reply 8: Thank you for spotting this error, it was corrected (“Thus, one has to keep in mind that results obtained with a certain questionnaire of interoception cannot be generalized to ISb.”)

______

Reviewers' comments:

Reviewer #1: The authors have addressed all of my comments regarding the manuscript. I recommend the paper for publication; however, I strongly suggest that the authors closely check the writing for grammar before going to press. There is also formatting inconsistency in use of decimal points or commas to separate the integer and fractional part of numbers.

Reply 9: Many thanks for the positive evaluation. The mentioned issues were addressed in the revised version.

Reviewer #3:

Final sentence is not clear – is the conclusion that we need a new questionnaire?

Reply 32: This is the second (more distal) part of the conclusion, which is preceded by a more direct conclusion (“These findings suggest that the investigated questionnaires cannot be used interchangeably to assess interoceptive sensibility.”). At the end of the discussion of the manuscript, we conclude that the existing questionnaires do have their weaknesses, thus there is room for another, theoretically well-founded questionnaire. ____

I didn’t have any concerns about the conclusion that the questionnaires should not be used interchangeably. You have provided data that support that conclusion. What is not clear is why you review the limitations of each questionnaire and conclude that a new questionnaire is needed. This is going beyond the aim of your study which was to see if the questionnaire measures the same underlying latent construct. If you want to critically review the interoception questionnaires, then this is a different aim and should be a different paper. I maintain that the final sentence of the abstract does not tally up to the aim of the study.

Reply 10: The final sentence recommending the development of a new questionnaire was removed from the conclusion.

Additionally, what was the reason for focusing on the noticing and ‘g’ scales of the MAIA? You mention in the introduction that the emotional awareness scale had previously been associated with the other questionnaire ……… yet you chose not to focus on the individual scales? This needs more justification. The ‘g’ scale seems to confound a range of different interoceptive constructs. I would recommend considering the scale separately – afterall, the aim of the present paper is to highlight to researchers that such differences need to be considered?

Reply 34: We think that the g-scale does not confound a range of different constructs but represents an overarching factor behind those constructs. In other words, the differentiation of the six subscales is not really justified, as shown by Ferentzi et al., 2020 (“However, a recent paper showed that the most prominent component of the construct can be well grasped with an overreaching factor (called MAIA-g in this paper) that includes six out of eight factors, while the remaining two (Not-Worrying and NotDistracting) are independent (Ferentzi et al., 2020).”). Thus, the use of the g-scale appears more appropriate than the inclusion of the separate scales. The only exception is the Noticing scale, because it measures primary experience (“Concerning the level of cognitive processing, only the Noticing subscale refers to the direct body experience, and, according to Mehling (2016), it is the most similar to the earlier questionnaires (and concepts) of bodily awareness.”). ____

I still disagree with this. There are now plenty of studies that find some sub-scales of the MAIA relate to particular disorders, while others don’t. This implies they are not measuring the same thing. I would like to see a factor analysis in supplementary information using the individual scales.

Reply 11: This analysis is presented as supplementary material in the revised version.

(3) My understanding of The Somatosensory Amplification Scale is that previous research has found it lacks internal reliability. One potential reason is that it may measure more than one construct. As teasing apart body related constructs was the aim of the present paper why use this scale to compare the others against?

Reply 35: We had two reasons. First, the SSAS was used in the validation process of the BPQ-BA and their positive association was interpreted as indicator of convergent validity. Second, somatosensory amplification is a construct that is very close to ISb (actually, it fits the definition of Garfinkel et al (2015, p.67): “the self-perceived dispositional tendency to be internally self-focused and interoceptively cognisant”; this is explicitly stated in the revised version), but usually does not considered an aspect of it, mainly because of its association with NA. This makes it an excellent anchor point for our study, which attempts to explore the associations between indicators of ISb and PA/NA. ____

This response doesn’t really address my question about reliability?

Reply 12: This issue was addressed in the limitations section as follows:“Fourth, the internal consistency of the SSAS was quite low. Cronbach’s alpha values in this domain are commonly reported in the literature and might reflect the conceptual heterogeneity of the construct (for a detailed discussion, see Köteles & Witthöft, 2017).”

ANALYSIS It might be problematic to add the noticing scale of the MAIA and ‘g’ together into the factor analysis when ‘g’ already contains the noticing scale………again this might be a reason to include each of the MAIA scales separately. Additionally, should the individual items, rather than the scales, go into the factor analysis?

Reply 36: The size of the sample does not allow the inclusion of individual items into the FA. To address the issue with the noticing subscale and the g-scale, we rerun the factor-analysis with a g-scale that does not contain the four items of the Noticing subscale (the correlation between the full and this shortened g-scale is .99). ____

Irrespective of the correlation it is still inappropriate to the noticing scale and the g scale (including noticing) into the same factor analysis as this would inflate the chances of the two loading onto the same factor. If removing noticing from the g scale does not influence the results then why not report that analysis? It would be more appropriate?

Reply 13: The factor analysis has been changed (see Reply 11). It does not contain the g-scale and does include all MAIA scales.

DISCUSSION The discussion is again very long and reads like a review of questionnaire measures rather than a discussion of the findings. While it raises a number of interesting points the focus should be on the present results. Reply 37: Similarly to the introduction, the discussion was substantially shortened in the revised version (see also Reply 28). ____ I am not sure I agree with the statements about an ideal interoception questionnaire L 459-464. Surely there is no such thing as an ‘ideal’ questionnaire, rather researchers should carefully select their questionnaire based on their research question and the underlying interoceptive construct they want to measure. I am not sure it makes conceptual sense to entirely separate interoception and affect as suggested in point 2 in this section. I am also not sure that an ‘ideal’ questionnaire would exclude the evaluative component – surely this is an important component, and again, whether it is assessed will depend on the research question. Reply 38: We think that the entire framework of Garfinkel and colleagues (2015) relies on the sensory aspect of interoception. We formulated our recommendations with keeping this approach in mind. This does not mean that the evaluative aspect is not important or can be completely separated from the sensory. This is elaborated in the discussion of the revised version.

The discussion still seems to go well beyond a discussion of the present results and the aim. Indeed, there are sections where your results are not referred to. In terms of Garfinkels definition, two scales recently developed by Murphy et al. seem to have taken care of that question. The Interoceptive Accuracy Scale and the Interoceptive Attention Scale. Please comment on what you think is needed beyond these two well validated and conceptualised scales.

Reply 14: Parts referring to the development of an ideal questionnaire were removed.

---

## [Decision Letter · Decision Letter 2]

10 May 2022

PONE-D-21-13357R2Questionnaires of interoception do not assess the same construct

PLOS ONE

Dear Dr. Vig,

Thank you for submitting your revised manuscript to PLOS ONE. Although one reviewer recommends acceptance at this stage, the other reviewer (reviewer 4) raises very important and valid concerns which I agree will need to be addressed before this paper could potentially be accepted. Therefore, we invite you to submit a revised version of the manuscript that addresses the points raised.

We look forward to receiving your revised manuscript.

Kind regards,

Jane Elizabeth Aspell, PhD

Academic Editor

PLOS ONE

Reviewers' comments:

Reviewer's Responses to Questions

**Comments to the Author**

1. If the authors have adequately addressed your comments raised in a previous round of review and you feel that this manuscript is now acceptable for publication, you may indicate that here to bypass the “Comments to the Author” section, enter your conflict of interest statement in the “Confidential to Editor” section, and submit your "Accept" recommendation.

Reviewer #1: All comments have been addressed

Reviewer #4: (No Response)

2. Is the manuscript technically sound, and do the data support the conclusions?

Reviewer #1: Yes

Reviewer #4: Partly

3. Has the statistical analysis been performed appropriately and rigorously? 

Reviewer #1: Yes

Reviewer #4: No

4. Have the authors made all data underlying the findings in their manuscript fully available?

Reviewer #1: Yes

Reviewer #4: Yes

5. Is the manuscript presented in an intelligible fashion and written in standard English?

Reviewer #1: Yes

Reviewer #4: No

6. Review Comments to the Author

Reviewer #1: I have no new comments. All of my requests from the previous round of revisions were addressed by the authors.

Reviewer #4: Thank you for the opportunity to review this research, particularly at this late stage in the peer-review process. Having read through the paper, and the dialogue between the authors and the other reviewers, I have some suggestions to add which may assist the authors in publishing the work, and making a useful contribution to the literature.

It is unfortunate timing perhaps, but I found the statement in the Introduction (and the premise of the paper) “Although the implicit assumption is that these questionnaires assess basically the same (or at least highly similar aspects of the same) phenomenon, this has not been systematically investigated to date” – to be incorrect. In fact, Desmedt, Heeren, and colleagues (2022) recently considered the associations between five self-report measures, including the three measures focused on in the present work (the BAQ, the BPQ and the MAIA). Given that Desmedt, Heeren, and colleagues (2022) used a much larger sample (n = 1003), a wider pool of measures, and more sophisticated set of analyses, I am not sure what the present work adds with regards to the associations between measures of interoception (particularly as the paper used a novel Hungarian translation of the BPQ, which has not been fully validated). The correlational analyses alone are not a strong assessment of the underlying associations between the measures.

To that end, my suggestion is that the authors refocus the paper in one of two ways:

• Focus on the reporting the validation of the Hungarian translation of the BPQ (see Swami & Barron, 2019 for reporting and analytic guidelines). I recommend that the authors collect more data if they pursue this option.

• Focus on the relationships between the interoception measures and the indices of positive and negative affect. For example, it would be interesting to see hierarchical regression analyses that identify the variance in positive/negative affect that is uniquely accounted for by each questionnaire measure. This could support the authors’ assertion that the questionnaire measures are incrementally distinct.

General

I suggest that the authors have the manuscript proofread for grammatical issues, which will enhance the clarity of the work.

Abstract

Line 1 – the phrasing of this sentence is not clear. I know you mean that questionnaire measures are sometimes referred to as interoceptive sensibility, but it looks as though you are saying that interoception itself is sometimes called interoceptive sensibility.

Introduction

The authors focus heavily on Garfinkel and colleagues’ (2015) tripartite model of interoception. However, more recent models (e.g., Desmedt, Luminet et al., 2022; Khalsa et al., 2018) might be a more appropriate and useful theoretical underpinning for distinguishing between the different components of interoception the authors purport to be measured by the BAQ, BPQ and MAIA.

Methods

Briefly comment on the I-PANAS-SF Hungarian valiation. i.e., did it support the original factor structure?

I recommend computing McDonald’s omega instead of Cronbach’s alpha for these questionnaire measures, as it performs more reliably.

Limitations

The present findings may be constrained by linguistic or local contextual factors, which might influence the understanding/meaning of the latent constructs, and potentially limit generalisability (for discussions, see Ma-Kellams, 2014; Todd, 2020; see also Swami & Barron, 2019).

Table 1

Why were some scores computed for some measures, and means for others? Consistency in computing means would facilitate comparison for readers across the measures.

References

Desmedt, O., Heeren, A., Corneille, O., & Luminet, O. (2022). What do measures of self-report interoception measure? Insights from a systematic review, latent factor analysis, and network approach. Biological Psychology, 108289. https://doi.org/10.1016/j.biopsycho.2022.108289

Desmedt, O., Luminet, O., Maurage, P., & Corneille, O. (2022). Discrepancies in the definition and measurement of interoception: A comprehensive discussion and suggested ways forward. https://doi.org/10.31234/osf.io/xd3nj

Ma-Kellams, C. (2014). Cross-cultural differences in somatic awareness and interoceptive accuracy: A review of the literature and directions for future research. Frontiers in Psychology, 5, 1379. https://doi.org/10.3389/fpsyg.2014.01379

Swami, V., & Barron, D. (2019). Translation and validation of body image instruments: Challenges, good practice guidelines, and reporting recommendations for test adaptation. Body Image, 31, 204-220. https://doi.org/10.1016/j.bodyim.2018.08.014

Todd, J., Barron, D., Aspell, J. E., Toh, E. K. L., Zahari, H. S., Khatib, N. A. M., & Swami, V. (2020). Translation and validation of a Bahasa Malaysia (Malay) version of the Multidimensional Assessment of Interoceptive Awareness (MAIA). PLoS One, 15(4), e0231048. https://doi.org/10.1371/journal.pone.0231048

7. PLOS authors have the option to publish the peer review history of their article (what does this mean?). If published, this will include your full peer review and any attached files.

Reviewer #1: No

Reviewer #4: No

---

## [Author Response · Author response to Decision Letter 2]

10 Jun 2022

Dear Dr. Aspell, dear Reviewers,

Thank you for the suggestions and comments on our manuscript. All issues you raised were carefully addressed in the revised version (see below). We believe, these changes considerably improved the overall quality of the manuscript. Hopefully, you will find the revised version appropriate for publication.

Best wishes,

Luca Vig,

corresponding author

Reviewer #4: Thank you for the opportunity to review this research, particularly at this late stage in the peer-review process. Having read through the paper, and the dialogue between the authors and the other reviewers, I have some suggestions to add which may assist the authors in publishing the work, and making a useful contribution to the literature.

It is unfortunate timing perhaps, but I found the statement in the Introduction (and the premise of the paper) “Although the implicit assumption is that these questionnaires assess basically the same (or at least highly similar aspects of the same) phenomenon, this has not been systematically investigated to date” – to be incorrect. In fact, Desmedt, Heeren, and colleagues (2022) recently considered the associations between five self-report measures, including the three measures focused on in the present work (the BAQ, the BPQ and the MAIA). Given that Desmedt, Heeren, and colleagues (2022) used a much larger sample (n = 1003), a wider pool of measures, and more sophisticated set of analyses, I am not sure what the present work adds with regards to the associations between measures of interoception (particularly as the paper used a novel Hungarian translation of the BPQ, which has not been fully validated). The correlational analyses alone are not a strong assessment of the underlying associations between the measures.

To that end, my suggestion is that the authors refocus the paper in one of two ways:

• Focus on the reporting the validation of the Hungarian translation of the BPQ (see Swami & Barron, 2019 for reporting and analytic guidelines). I recommend that the authors collect more data if they pursue this option.

• Focus on the relationships between the interoception measures and the indices of positive and negative affect. For example, it would be interesting to see hierarchical regression analyses that identify the variance in positive/negative affect that is uniquely accounted for by each questionnaire measure. This could support the authors’ assertion that the questionnaire measures are incrementally distinct.

Reply 1 We don't think that timing (which often depends on the availability of the reviewers, as in the case of this paper) should be a concern here. Our study has been planned well before the mentioned article was published. The fact that two different studies using different samples and analyses has similar results underlines the validity of them both. We believe even if one single empirical study (not without weak points) found something, it is still worth investigating the same question using another sample and method. In the era of open science, we think it should be ok to have similar papers published a few months apart.

As for the practical part, we focus a bit more on the associations between the assessed indicators of ISb and positive and negative affect (including regression).

General

I suggest that the authors have the manuscript proofread for grammatical issues, which will enhance the clarity of the work.

Reply 2 The manuscript has been carefully proofread.

Abstract

Line 1 – the phrasing of this sentence is not clear. I know you mean that questionnaire measures are sometimes referred to as interoceptive sensibility, but it looks as though you are saying that interoception itself is sometimes called interoceptive sensibility.

Reply 3 Thank you for pointing out this issue, it was corrected. („There are a number of questionnaires assessing the self-reported trait-like aspect of interoception, also called interoceptive sensibility (ISb).”)

Introduction

The authors focus heavily on Garfinkel and colleagues’ (2015) tripartite model of interoception. However, more recent models (e.g., Desmedt, Luminet et al., 2022; Khalsa et al., 2018) might be a more appropriate and useful theoretical underpinning for distinguishing between the different components of interoception the authors purport to be measured by the BAQ, BPQ and MAIA.

Reply 4 The intorduction was extended with more recent models. (see the definition (first paragraph of the Introduction) and the refinement of the taxonomy following the model of Garfinkel and colleagues (2015) (second paragraph of the Introduction))

Methods

Briefly comment on the I-PANAS-SF Hungarian valiation. i.e., did it support the original factor structure?

I recommend computing McDonald’s omega instead of Cronbach’s alpha for these questionnaire measures, as it performs more reliably.

Reply 5 Factor structure of the used version of the PANAS was confirmed by CFA (Gyollai et al., 2011). In the revised version, McDonald’s omega is used as indicator of internal consistency.

Limitations

The present findings may be constrained by linguistic or local contextual factors, which might influence the understanding/meaning of the latent constructs, and potentially limit generalisability (for discussions, see Ma-Kellams, 2014; Todd, 2020; see also Swami & Barron, 2019).

Reply 6 We added this point to the limitations. („Fifth, cultural and linguistic factors might also limit the generalizability of the findings (for discussion see Ma-Kellams, 2014).”)

Table 1

Why were some scores computed for some measures, and means for others? Consistency in computing means would facilitate comparison for readers across the measures.

Reply 7 We always followed the formula used by the authors of the original questionnaire. This is always useful if one want to compare findings with those of other studies. As the present study is interested in associations between constructs, differences in the calculation of total scores (i. e. sum of item scores or average) do not impact the results.

---

## [Decision Letter · Decision Letter 3]

12 Jul 2022

PONE-D-21-13357R3Questionnaires of interoception do not assess the same constructPLOS ONE

Dear Dr. Vig,

Thank you for submitting your manuscript to PLOS ONE. One reviewer has  recommended the paper be accepted but the other reviewer has a few minor remaining issues that need addressing.

We look forward to receiving your revised manuscript.

Kind regards,

Jane Elizabeth Aspell, PhD

Academic Editor

PLOS ONE

Journal Requirements:

Reviewers' comments:

Reviewer's Responses to Questions

**Comments to the Author**

1. If the authors have adequately addressed your comments raised in a previous round of review and you feel that this manuscript is now acceptable for publication, you may indicate that here to bypass the “Comments to the Author” section, enter your conflict of interest statement in the “Confidential to Editor” section, and submit your "Accept" recommendation.

Reviewer #4: (No Response)

2. Is the manuscript technically sound, and do the data support the conclusions?

Reviewer #4: Yes

3. Has the statistical analysis been performed appropriately and rigorously? 

Reviewer #4: Yes

4. Have the authors made all data underlying the findings in their manuscript fully available?

Reviewer #4: Yes

5. Is the manuscript presented in an intelligible fashion and written in standard English?

Reviewer #4: Yes

6. Review Comments to the Author

Reviewer #4: Thank you for the opportunity to review this work again. The authors have addressed most of my comments thoroughly. I have just a few remaining minor suggestions, following which I believe the work will be suitable for publication.

I refer to the editor for guidance on PLos referencing style, which appears incorrect.

Participants:

- Did participants receive any payment?

- What was deemed to be a ‘high proportion of missing items’? e.g., >20%?

- Is data available for additional participant characteristics that would better characterise the representativeness of the sample, E.g., occupation status etc.?

Line 392 tacked changes document, ‘details’ should be changed back to ‘detail’

Supplementary materials

In line with my previous review comments, I have some concerns about the novel psychometric validation of the BPQ (I recommend further data collection for a full validation, if possible).

- The authors do not indicate if or how they divided the sample for the EFA and CFA analyses (it is not appropriate to conduct EFAs and CFAs using the same sample). See Worthington & Whittaker (2006) or Swami and Barron (2019) for sample size recommendations for EFA based on item communalities. Given the small sample, I recommend prioritising EFA over CFA (see Swami et al., 2021), and recommending that CFA is performed in the future in the Discussion.

- Principal axis would be preferable to maximum likelihood (see Watkins, 2018; Statistical simulations have found that PA outperforms ML when the relationships between measured variables and factors are relatively weak (≤.40), sample size is relatively small (≤300), multivariate normality is violated, or when the number of factors underlying the measured variables is misspecified)

- What rotation was used for the EFA? (e.g., Quartimax, given the expectation of a single orthogonal factor?)

- How did you determine the number of factors to extract in the EFA? (e.g., eigenvalues >1, parallel analysis, examination of scree plot, minimum average partials, etc.?) I recommend parallel analysis or MAP

7. PLOS authors have the option to publish the peer review history of their article (what does this mean?). If published, this will include your full peer review and any attached files.

Reviewer #4: No

---

## [Author Response · Author response to Decision Letter 3]

28 Jul 2022

Dear Dr. Aspell, dear Reviewers,

Thank you for all your comments and suggestions. We addressed the issues raised; hopefully, you will find the revised version appropriate for publication.

Best wishes,

Luca Vig,

corresponding author

Reviewer #4: Thank you for the opportunity to review this work again. The authors have addressed most of my comments thoroughly. I have just a few remaining minor suggestions, following which I believe the work will be suitable for publication.

I refer to the editor for guidance on PLos referencing style, which appears incorrect.

Reply 1 Thank you for noticing this mistake, we have changed the reference style from APA 7th edition to Vancouver style using Zotero. We checked and made sure that the reference list is complete and correct as requested. To our knowledge, none of the cited articles have been retracted.

Participants:

- Did participants receive any payment?

Reply 2 We added this information („Participants did not receive any payment; but they could receive feedback on their scores of the eight MAIA subscales and their explanation based on the original publication of the questionnaire (Mehling et al., 2012) following the data collection period if they requested.”

- What was deemed to be a ‘high proportion of missing items’? e.g., >20%?

Reply 3 We specified this issue. (“Originally, 392 individuals started to fill out the questionnaire; 127 of them were excluded because they quit before completing the entire test battery.”)

- Is data available for additional participant characteristics that would better characterise the representativeness of the sample, E.g., occupation status etc.?

Reply 4 We added information about the level of education. (“In terms of educational qualification, 78,5% reported higher education (university degree), 20,8 % secondary level education (high school), and less than 1 % primary level education.”)

Line 392 tacked changes document, ‘details’ should be changed back to ‘detail’

Reply 5 Thank you for pointing out this mistake, we corrected it.

Supplementary materials

In line with my previous review comments, I have some concerns about the novel psychometric validation of the BPQ (I recommend further data collection for a full validation, if possible).

- The authors do not indicate if or how they divided the sample for the EFA and CFA analyses (it is not appropriate to conduct EFAs and CFAs using the same sample). See Worthington & Whittaker (2006) or Swami and Barron (2019) for sample size recommendations for EFA based on item communalities. Given the small sample, I recommend prioritising EFA over CFA (see Swami et al., 2021), and recommending that CFA is performed in the future in the Discussion.

- Principal axis would be preferable to maximum likelihood (see Watkins, 2018; Statistical simulations have found that PA outperforms ML when the relationships between measured variables and factors are relatively weak (≤.40), sample size is relatively small (≤300), multivariate normality is violated, or when the number of factors underlying the measured variables is misspecified)

- What rotation was used for the EFA? (e.g., Quartimax, given the expectation of a single orthogonal factor?)

- How did you determine the number of factors to extract in the EFA? (e.g., eigenvalues >1, parallel analysis, examination of scree plot, minimum average partials, etc.?) I recommend parallel analysis or MAP

Reply 6: CFA was removed, EFA was performed using your recommendations. The text was corrected as follows: “To explore the factor structure of BPQ, exploratory factor analysis using principal axis factoring with quartimax rotation was performed. The data set was appropriate for factor analysis (Kaiser-Meyer-Olkin test with values > 0.9 showed adequate sampling; Barlett-test of sphericity was significant). Parallel analysis indicated a single factor; also, only the first factor had an eigenvalue above 1 (14.394), This factor explained 55.4% of the total variance. All items loaded above 0.5 on this factor; factor loadings are presented in S1 Table.”

---

## [Decision Letter · Decision Letter 4]

8 Aug 2022

Questionnaires of interoception do not assess the same construct

PONE-D-21-13357R4

Dear Dr. Vig,

We’re pleased to inform you that your manuscript has been judged scientifically suitable for publication and will be formally accepted for publication once it meets all outstanding technical requirements.

Kind regards,

Jane Elizabeth Aspell, PhD

Academic Editor

PLOS ONE

Reviewers' comments:

Reviewer's Responses to Questions

**Comments to the Author**

1. If the authors have adequately addressed your comments raised in a previous round of review and you feel that this manuscript is now acceptable for publication, you may indicate that here to bypass the “Comments to the Author” section, enter your conflict of interest statement in the “Confidential to Editor” section, and submit your "Accept" recommendation.

Reviewer #4: All comments have been addressed

2. Is the manuscript technically sound, and do the data support the conclusions?

Reviewer #4: Yes

3. Has the statistical analysis been performed appropriately and rigorously? 

Reviewer #4: Yes

4. Have the authors made all data underlying the findings in their manuscript fully available?

Reviewer #4: Yes

5. Is the manuscript presented in an intelligible fashion and written in standard English?

Reviewer #4: Yes

6. Review Comments to the Author

Reviewer #4: The authors have fully addressed all of my previous comments and I am happy to recommend publication.

7. PLOS authors have the option to publish the peer review history of their article (what does this mean?). If published, this will include your full peer review and any attached files.

Reviewer #4: No

---

## [Editor Report · Acceptance letter]

12 Aug 2022

PONE-D-21-13357R4 

Questionnaires of interoception do not assess the same construct 

Dear Dr. Vig:

I'm pleased to inform you that your manuscript has been deemed suitable for publication in PLOS ONE. Congratulations! Your manuscript is now with our production department. 

Kind regards, 

on behalf of

Dr. Jane Elizabeth Aspell 

Academic Editor

PLOS ONE